# Novel Oleanolic Acid-Phtalimidines Tethered 1,2,3 Triazole Hybrids as Promising Antibacterial Agents: Design, Synthesis, In Vitro Experiments and In Silico Docking Studies

**DOI:** 10.3390/molecules28124655

**Published:** 2023-06-08

**Authors:** Ghofrane Lahmadi, Mabrouk Horchani, Amal Dbeibia, Abdelkarim Mahdhi, Anis Romdhane, Ata Martin Lawson, Adam Daïch, Abdel Halim Harrath, Hichem Ben Jannet, Mohamed Othman

**Affiliations:** 1Normandie University, URCOM, UNILEHAVRE, FR3021, UR 3221, 25 Rue Philippe Lebon, BP 540, F-76058 Le Havre, France; ghofrane.lahmadi@etu.univ-lehavre.fr (G.L.); lawsona@univ-lehavre.fr (A.M.L.); adam.daich@univ-lehavre.fr (A.D.); 2Laboratory of Heterocyclic Chemistry, LR11ES39, Faculty of Science of Monastir, University of Monastir, Avenue of Environment, Monastir 5019, Tunisia; horchani.mabrouk@gmail.com (M.H.); anis_romdhane@yahoo.fr (A.R.); 3Laboratory of Analysis, Treatment and Valorization of Pollutants of the Environment and Products, Faculty of Pharmacy, University of Monastir, Monastir 5000, Tunisia; amaldbeibia@gmail.com (A.D.); abdelkarim.mah@gmail.com (A.M.); 4Department of Zoology, College of Science, King Saud University, Riyadh 11451, Saudi Arabia; hharrath@ksu.edu.sa

**Keywords:** oleanolic acid, phtalimidines, triazole, click chemistry, antibacterial activity, molecular docking

## Abstract

As part of the valorization of agricultural waste into bioactive compounds, a series of structurally novel oleanolic acid ((3β-hydroxyolean-12-en-28-oic acid, **OA-1**)-phtalimidines (isoindolinones) conjugates **18a**–**u** bearing 1,2,3-triazole moieties were designed and synthesized by treating an azide **4** previously prepared from **OA-1** isolated from olive pomace (*Olea europaea* L.) with a wide range of propargylated phtalimidines using the Cu(I)-catalyzed click chemistry approach. **OA-1** and its newly prepared analogues, **18a**–**u,** were screened in vitro for their antibacterial activity against two Gram-positive bacteria, *Staphylococcus aureus* and *Listeria monocytogenes*, and two Gram-negative bacteria, *Salmonella thyphimurium* and *Pseudomonas aeruginosa*. Attractive results were obtained, notably against *L. monocytogenes*. Compounds **18d**, **18g**, and **18h** exhibited the highest antibacterial activity when compared with **OA-1** and other compounds in the series against tested pathogenic bacterial strains. A molecular docking study was performed to explore the binding mode of the most active derivatives into the active site of the ABC substrate-binding protein Lmo0181 from *L. monocytogenes*. Results showed the importance of both hydrogen bonding and hydrophobic interactions with the target protein and are in favor of the experimental data.

## 1. Introduction

Agricultural waste has emerged as a huge pool of fine chemicals that can be turned into high-value compounds with many pharmaceutical applications [1]. Triterpenic acids, such as Oleanolic Acid (**OA-1**, Figure 1) extracted from olive pomace [2], are typical examples of such high-value compounds. A number of studies have demonstrated that **OA-1** has a wide range of biological activities, including anti-inflammatory [3], hepatoprotective [4], antioxidant [5], antitumor [6,7], anti-HIV [8], antidiabetic [9], and antiparasitic [10] effects. It has also been found that **OA-1** and its derivatives have significant antibacterial activity with a wide range of MIC values [11,12,13]. The antibacterial activity of **OA-1** is thought to be due to its ability to influence peptidoglycan structure and composition, gene expression, and biofilm formation, thereby preventing bacterial growth [14]. On the other hand, Nitrogen-containing heterocyclic compounds are omnipresent in bioactive molecules and are invaluable sources of therapeutic agents [15,16,17]. Among them, Phtalimidine (1-isoindolinone and C3 1-isoindolinone-derived) derivatives [18] play an important role in drug discovery due to their broad and abundant biological activities [19,20,21].Many of them have been prepared and examined as antihypertensive [22], antipsychotic [23,24,25], anti-inflammatory [26], anesthetic [27], and vasodilatory [28] agents. Antiviral [29,30,31,32], anticancer [33,34,35,36], antimicrobial [37,38], and anxiolytic [39] activities have also been observed in this class of structures (Figure 1). As a result, a lot of study has been completedover the past few decades to synthesize and to explore various therapeutic prospective of this moiety [18,40,41].

Despite its multiple potential applications, **OA-1** has not yet been developed as a medicine due to its instability and limited water solubility. To overcome these drawbacks, several studies have been designed to modify the structure of **OA-1** with the hope of improving its physical properties to obtain better bioavailability and higher bioactivity. Among the plethora of strategies used, molecular hybridization, which is a new concept in drug design and development based on the combination of two different bioactive compounds to produce a new hybrid substance, has emerged as an essential tool for design and construction of novel hybrid molecules with improved biological activities [42,43,44,45,46,47,48]. Given the rising incidence of multidrug-resistant pathogens caused by the widespread use of antibiotics [49], the intriguing biological activities of isoindolinones and **OA-1**, and our current research interest in the valorization of agricultural waste into eco-efficient, bioactive products, we present here the synthesis of novel triazole-tethered isoindolinones oleanolic acid hybrids by means of click chemistry-mediated fusion between isoindolinones and oleanolic acid derivatives. To that end, a Cu(I)-catalyzed azide alkyne Huisgen 1,3-cycloaddition was used [50,51,52,53]. Moreover, we hope that the introduction of triazole linkers, known by their broad and abundant biological properties (antiviral [54], antioxidant [55,56], antimicrobial [57], anticancer [58,59], antimalarial [60,61]...) could contribute to the improvement of the overall biological activity of the hybrid molecules [62,63,64,65,66]. The antibacterial activity of **OA-1** and the target compounds werescreened, followed by in silico molecular docking studies for the most potent derivatives, in order to obtaina better understanding about the interactions and binding mode in the active sites of the target protein.

## 2. Results and Discussion

### 2.1. Chemistry

#### 2.1.1. Isolation of Oleanolic Acid OA-1

**OA-1** (Figure 1) was isolated from olive pomace (*Olea. europaea* L.) cultivar, Chemlali, by using a solid–liquid and ultrasound-assisted extraction strategy, as previously described by us [67]. This method is low-cost, selective, and provides a large amount of **OA-1** of around 6.8 g (3.4 mg/g DW)

#### 2.1.2. Synthesis

The new hybrid molecules **18** were designed to include **OA-1** on one hand and isoindolinones **6**–**8** and (±)-**12**–**17** on the other, by connecting them via linker chains of different length. The Cu(I)-catalyzed azide–alkyne cycloaddition (CuAAC) was chosen as the linking methodology. Starting from the naturally occurring triterpene **OA-1**, azidoacetyl **4,** as the first precursor, was synthesized in three steps according to a previously reported procedure (Figure 1) [68,69].

The second precursors, namely N/O-propargylated isoindolinones **6**–**8,** were also synthesized using previously published protocols [70,71,72,73,74,75,76]. Substrates **6a**–**g** were prepared through two synthetic routes involving base-assisted N-alkylation of phtalimide **5a**, hydantoin **5b**, succinimide **5c,** and saccharin **5d** rings with propargyl bromide resulting in the isolation of the precursors **6a**–**d** in well isolated yields (87 to 97%). Under the same conditions, deprotonation and then alkylation of N-hydroxyphthalimide**5g** afforded compound **6g** in 67% isolated yield. Applied to isoindolinone **5a** and successively but-3-yn-1-ol or but-3-yn-2-ol, the mitsunobu reaction [77] (Ph_3_P/DEAD/THF) provided isoindolinones **6e** and **6f** in 97% and 57% isolated yields, respectively (Figure 2). Wishing to study the effect of the presence of a hydroxy and or acetoxy group on the antibacterial activity of our molecules, we subsequently prepared the hydroxylactams **7a** and **7e,** as well as the acetoxylactam **8g**. In this sense, the selective reduction of propargylatedisoindolinones **6a** and **6e** was performed by using excess of NaBH_4_ (4 equiv) in a dry MeOH at 0 °C [78] leading to the hydroxylactams **7a** and **7e** in 95% and 50% isolated yields, respectively (Figure 2). Using our recently reported one-pot reduction–acetylation protocol, imide **6g** afforded the α-acetoxy-lactam **8g** in a 98% overall yield (Figure 2) [79].

Subsequently, phtalimidines(±)-**12**–**17**, the other precursors for the Cu(I)-catalyzed Huisgen 1,3-dipolar cycloaddition reaction, were obtained straightforwardly, starting with homophthalic acid (**HPA**-**9**), as previously described by our group [80,81,82]. Phtalimidines (±)-**12a**–**f**, as starting materials, were obtained by using our well-known 3-step sequence, including (i) esterification of diacid (**HPA**-**9**) under reflux in the presence of gaseous HCl, (ii) radical bromination of the obtained diester **10** (e.g., NBS, AIBN_cat_, CCl_4_ at reflux), and(iii) condensation of excess of primary amine (α-bromophthalate**11**, R-NH_2_, CH_3_CN, rt). Next, the C3-alkylated derivatives (±)-**13** and (±)-**14**, were prepared by the deprotonation of the α-position of the nitrogen of phtalimidines (±)-**12a**–**f** with potassium carbonate, followed by the reaction of various halogenated electrophiles. Under these conditions, substrates (±)-**13a**–**d** and (±)-**14b**–**f**, were obtained in good to excellent isolated yields 87% to 90% for (±)-**13a**–**d** and 81% to 93% for (±)-**14b**–**f** (Figure 3).

In order to evaluate the binding mode, particularly the effect of an hydroxy, amide, or carboxy group at C-3, on the antibacterial activity, acid (±)-**15f**, amide (±)-**16b** and alcohol (±)-**17e** were then prepared. The reduction of the ester group at C-3 of (±)-**14e** with LiBH_4_ in dichloromethane at room temperature gave phtalimidine alcohol (±)-**17e** in 80% isolated yield. Saponification of ester functions of esters (±)-**14b** and (±)-**14f** under standars conditions (excess of aqueous NaOH, EtOH, rt then diluted HCl at 0 °C) gave the corresponding carboxylic acids (±)-**15b** and (±)-**15f** in, respectively, 93% and 72% isolated yields. In order to obtain amide (±)-**16b**, acid (±)-**15b** was treated, in a next step, with methyl glycinate in the presence of EDCI/DMF at room temperature leading to compound (±)-**16b** in 75% isolated yield (Figure 3) [83]. With azide **4** and propargylated phthalimidines **6**–**8** and (±)-**12**–**17** in hand, we next directed our attention to explore the 1,3-dipolar cycloaddition to form the new hybrid molecules **18**. Thus, as shown in Figure 4, treatment of **4** and **6a** used as a model for our study under the Cu(I)-catalyzed Huisgen 1,3-dipolar cycloaddition (e.g., CuSO_4_.5H_2_O, sodium ascorbate (NaC_6_H_7_O_6_), DCM/H_2_O, rt 24 h) [84] resulted in the formation of the 1,2,3-triazole conjugate **18a** in 84% isolated yield.

The above selected conditions were used with the rest of the substrates **6**–**8** and (±)-**12**–**17**, which cyclized to afford the hybrids molecules **18b**–**t** with yields of 70 to 98% after silica gel column chromatography, and the results are summarized in Figure 5.

It is worth noting that in the presence of the C,N-bispropargylatedphtalimidine (±)-**13d** under standard conditions, the 1,3-dipolar cycloaddition became non-selective and lead to a mixture of non-separable products, including the mono-cycloaddition product on the C-propargylated alkyne, the mono-cycloaddition on the N-propargylated alkyne and the bis cycloadduct **18u**. After rigorous investigations, it was observed that the double cycloaddition occurred seamlessly to give **18u** in 80% yield when CuSO_4_.5H_2_O (0.2 equiv), sodium ascorbate (0.4 equiv), and 2 equivalents of **4** were used (Figure 6).

### 2.2. Antibacterial Activity

All the newly synthesized compounds were evaluated in vitro for their antibacterial activity against the following human pathogen strains, two Gram-positive bacteria, *Staphylococcus aureus ATCC 25923* and *Listeria monocytogenes ATCC 19115*, and two Gram-negative bacteria, *Salmonella thyphimurium ATCC 14080* and *Pseudomonas aeruginosa ATCC 27853*. The activity of all the tested derivatives was compared with that of Tetracycline and Chlorhexidine used as standard reference antibiotics. The determination of the inhibition zone “IZ” (in mm), MIC and MBC (in µM) was carried out in this study. Overall, attractive results were obtained against certain strains. Indeed, the values of the IZ diameters, as a preliminary test, given in Table 1 for certain compounds were found to be in agreement with those of the MIC and MBC describedin Table 2. These results showed that the starting product **OA-1** was active against *S. aureus*, *S. thyphimurium,* and *P. aeruginosa,* but inactive towards *L. monocytogenes*. In certain circumstances, the activity of this compound exceeds that of specific derivatives and reference antibiotics. It was discovered to be more active against *S. aureus* and *P. aeruginosa* than all of its derivatives **18a**–**u**, but only slightly more active against *S. typhimurium* than **18f**, **18k**, and **18q**. The results showed, on the other hand, that most of the tested compounds demonstrated a certain level of selectivity towards *L. monocytogenes*. For more details, the most interesting results in terms of MIC were noted with the derivatives **18a**–**h** in addition to **18k**, **18m,** and **18q** which showed good inhibitory effects on the growth of *L. monocytogenes*, compared to that of the reference antibiotics **TET** (MIC = 576.01 µM) and **CHX** (MIC = 253.24 µM). Thus, from a structural point of view, the compound **18g** exhibits the highest activity (9.48 µM) towards this Gram-positive strain compared to the rest of the active compounds followed by derivatives **18d** (9.56 µM) and **18h** (9.89 µM). The observed high antibacterial activity of these compounds may be attributed to their sulfonyl and hydroxyl functional groups, as well as specific arrangements within their structures that enhance molecular interactions, binding affinity, or target recognition with this Gram-positive bacterium. It appears that the 3-hydroxyisoindolin-1-one fragment in compound **18h** contributes to this activity compared to its analog **18a** with an isoindoline-1,3-dione moiety (MIC = 12.4 μM). This finding shows that the additional hydroxyl group in **18h** instead of the carbonyl function in **18a** may account for the noted difference in activity. On the other hand, the higher activity of compound **18a** against *L. monocytogenes* (Table 2), compared to that of its analog **18e** allows to conclude that a single methylene binding the phthalimide to the trizole was better than two methylenes giving, perhaps, to this system freer rotation and, therefore, less possibility of interaction with the target proteins of the bacterium. Additionally, we noticed that when the methylene linker directly bonded to the phthalimide nitrogen atom in **18e** (MIC = 39.01 µM and MBC = 2496.67 µM) is replaced by an oxygen atom in **18g** (MIC = 9.48 µM and MBC = 155.65 µM), the antibacterial potential towards *L. monocytogenes* was much improved, hence the importance of this new linker (-O-CH_2_-) between phatlimide and triazole to better inhibit this strain. In addition, branching of the ethyl linker in **18e** was found to significantly improve the MBC of its analog **18f** (MBC = 624.16 µM). The additional methyl group in **18f** compared to **18a**, more than doubled the activity. Moreover, we noticed that compound **18a** with an isoindoline-1,3-dione moiety (MIC = 12.4 µM) remains more active against *L. monocytogenes* than its analogue **18c** with pyrrolidine-2,5-dione moiety (MIC = 21.13 µM). This finding serves as evidence that the antibacterial potential is affected by the additional aromatic ring. Interestingly, against *P. aeruginosa*, compound **18m** exhibited the highest antibacterial activity (MIC = 42.79 µM and MBC = 171.18 µM) compared to the other active compound **18h** (MIC = 158.41 µM and MBC = 633.66 µM) and also to the reference antibiotics. The results discussed above demonstrated the assumptions about the structural activity relationship (SAR) and also can be supported with some in silico studies.

### 2.3. Molecular Docking Study

The molecular docking simulations applied to antibacterial agents tested in vitro gives a descriptive explanation of the ligand’s binding mode for the inhibition of the target bacterium [85,86].

The results of the in vitro antibacterial test which showed the sensitivity of *L. monocytogenes* for the synthesized compounds, prompted us to obtainfurther insight about the inhibitory effect involving the active site. Moreover, previous studies have provided significant information on the function of the cycloalternan pathway (cycloalternane: ligand co-complexed with ABC substrate-binding protein Lmo0181) and revealed the mechanism of regulators of the transcription repressor, open reading frame kinase (ROK). These studies, which have developed a structural overview, allow us to anticipate the role of the cycloalternan (CA) pathway in the metabolism of starch derivatives and prove its involvement in the pathogenesis of the ABC substrate-binding protein from *L. monocytogenes*. They suggest that CA plays a role in interspecific competition for resources potentially in the host’s gastrointestinal tract and create the methodological framework for characterizing bacterial systems of unknown function [87]. All these data motivated us to choose ‘ABC substrate-binding protein Lmo0181′ as the target receptor of the docking-complex in order to predict the inhibitory effect of compounds docked in the CA’s active site.

In this context, to pick up the mode of action of the tested derivatives for their antibacterial potentials, the molecular docking study has been used to determine the binding modes against ABC substrate-binding protein Lmo0181 from *L. monocytogenes* (PDB ID: 5F7V). As depicted in Table 3, the listed binding affinities of the formed complex were found to be in the range of −12.3 to −10.6 kcal/mol. Thus, from these results, it can be suggested that all tested compounds interact favorably with the target protein and especially for derivatives **18c**, **18d**, **18h,** and **18k** which showed the best binding scores (−12.3 to −11.6 kcal/mol) even better than the docked antibiotic “Tetracycline” (−11.1 kcal/mol) used as a reference in the in vitro test. Interestingly, as Figure 2 and Figure 3A–D show, the binding modes for these compounds demonstrate that each ligand was located inside the binding cavity, similar to the co-crystalized inhibitor. Therefore, the best docking score of compound **18h** could be attributed to its correct orientation in the receptor cavity (Figure 3A) and it depends on its structure containing the 3-hydroxyisoindolin-1-one moiety, which could contribute to the stability of the receptor-ligand complex by the formation of three intermolecular conventional hydrogen bonds, two by its hydroxyl group with the residues Asn380 (2.55 Å) and Asp384 (2.12 Å), and another one by its carbonyl function with Thr75 (3.13 Å) amino acid (Figure 4C′). In addition, the stability of the complex is also perceptible through other hydrophobic interactions formed via the hydrocarbon skeleton of the molecule: Alkyl, Pi-Alkyl, and Pi-Pi as depicted in Table 4. The second best score was designated to compound **18d** with −12.0 kcal/mol which is also involved in three conventional H-bonds formed through its benzo[d]isothiazol-3(2H)-one 1,1-dioxide fragment with Glu391 (3.27 Å) (3.38 Å) and its ester function with Trp271 (3.15 Å). As docking results of compound **18h**, the derivative **18d** also displayed some hydrophobic interactions (Figure 4B′). On another hand, besides other types of interactions, the ligand **18c** (−11.7 kcal/mol) showed a single hydrogen bond through the carbonyl function of its pyrrolidine-2,5-dione pharmacophore with Trp271 (3.13 Å) (Figure 4A′), while the derivative **18k** (−11.6 kcal/mol), besides forming an hydrogen bond with Trp271 (3.25 Å), displayed diverse interactions which additionally explain the stability of the docking complex, as well as its inhibitory effect towards *L. monocytogenes* (Figure 4D′). Attempt to validate QSAR model by using docking results demonstrated a high degree of correlation in terms of the ability to form hydrogen bonds, to display good binding scores and the antibacterial potentials of our synthesized compounds.

## 3. Materials and Methods

### 3.1. General Experimental Procedure

Commercially available compounds were used without further purification (suppliers: Thermo Fisher Scientific Inc., and Sigma-Aldrich Co. (Asnières-sur-Seine and Saint-Quentin-Fallavier, France)). All glass apparatus was oven dried and cooled under vacuum before use. Before their usage, precautions were taken to eliminate moisture by refluxing over CaH_2_ while distilling the solvents (CH_3_CN, CH_2_Cl_2_). Column chromatographies were carried out with a BUCHI Pure Flash/Prep C-850 chromatography system using puriFlash^®^ packed columns. Distilled solvents were employed as eluents for column chromatography in all cases. Thin layer chromatographies (TLC) wererealized on sheets of silica gel 60 precoated with fluorescent indicator UV254 (Merck). Detection was performed by irradiation with a UV lamp and by using an ethanolic solution of *p*-anisaldehyde. Melting points were measured using a Stuart Scientific SMP10 apparatus and are uncorrected. IR spectra were recorded on a Perkin-Elmer FT-IR Paragon 1000 spectrometer. ^1^H NMR and ^13^C NMR spectra were recorded on a Bruker AvanceIIITM 300 MHz spectrometer at room temperature (rt) with tetramethylsilane (TMS) serving as internal standard. Chemical shifts are expressed in parts per million (δ). Splitting patterns are designed, s, singlet; d, doublet; dd, doublet of doublets; t, triplet; m, multiplet; and br. s, broaden singlet. Employed abbreviations refers to Ph: Phenyl, Trz: Triazole, Hyd: Hydantoin, and Fur: Furane. Coupling constants (*J*) are reported in Hertz (Hz). Mass spectra (GC-MS) were obtained on a ThermoFinniganAutomass III spectrometer coupled with a gas chromatograph Trace GC 2000. An agilent 6530 Q-Tof MS system was used to conduct the measurement of high-resolution mass spectra (HRMS).

### 3.2. Chemistry

#### General Procedure for the Preparation of Compounds **18**

CuSO_4_.5H_2_O (0.2 equiv.) and sodium ascorbate (0.4 equiv) were added to a mixture of equimolar amounts of azide 4 and propargylated phtalimidines **6**–**8** and (±)-**12**–**17** in CH_2_Cl_2_ (1 mL) and H_2_O (1 mL). After stirring at room temperature for 24h, the resulting residue was concentrated under vacuum conditions and then extracted with CH_2_Cl_2_ (3 × 10 mL). The combined organic layers were washed with brine, dried using MgSO_4_, filtered, and concentrated. The residue was then purified by flash column chromatography using a mixture of cyclohexane/EtOAc as eluent.

The structures of all the synthesized compounds **18** were confirmed by spectroscopic analysis. For example, the IR spectrum of **18e** shows a sharp intense band at 1714 cm^−1^ attributed to the four C=O functions. The ^1^H NMR spectrum of the same compound shows a singlet at δ_H_ 7.56 corresponding to the triazole proton and two triplets at δ_H_ 4.02 and 3.17 (J = 7.3 Hz) corresponding to the two methylene groups at the junction of the phthalimide and the triazole moities. The ^13^C NMR spectrum of **18e** shows two characteristic signals at δ_C_ 177.5 and 168.2 corresponding to the C=O of the ester and the acetoxy groups, respectively, and an additional signal at δ_C_ 166.1 attributable to the two C=O moieties of the Phtalimide group was also observed.

*(4aS,6aS,6bR,8aR,10S,12aR,12bR,14bS)-benzyl 10-(2-(4-((1,3-dioxoisoindolin-2-yl)methyl)-1H-1,2,3-triazol-1-yl)acetoxy)-2,2,6a,6b,9,9,12a-heptamethyl-1,2,3,4,4a,5,6,6a,6b,7,8,8a,9,10,11,12,12a,12b,13,14b-icosahydropicene-4a-carboxylate* (**18a**), This derivative was isolated as a white solid in 84% yield; Rf (cyclohexane/EtOAc: 60/40) = 0.5; mp = 84–86 °C; [α]_D_^20^ +57 (c 0.95 mg/mL, CH_2_Cl_2_); **IR (ν_max_/cm^−1^)** 2922.91 (CH str.), 1716.51 (4 × C=O); **^1^H NMR (300 MHz, CDCl_3_)**δ_H_ 7.90–7.83 (m, 2H, 2 × CH_aro_), 7.79–7.71 (m, 3H_,_ 3 × CH_aro_), 7.41–7.30 (m, 5H, 5 × CH_aro_), 5.29 (t, *J* = 3.8 Hz, 1H, CH=C), 5.15–5.02 (m, 6H, NCH_2_Trz, TrzCH_2_CO_2_, OCH_2_Ph), 4.54 (dd, *J* = 10.8, 5.2 Hz, 1H, CHCO_2_), 2.91 (dd, *J* = 14.2, 4.4 Hz, 1H, C=CC**H**CH_2_), 2.05–1.72 (m, 4H, 2 × CH_2_), 1.70–1.55 (m, 8H, 4 × CH_2_), 1.50–1.33 (m, 4H, 2 × CH_2_), 1.30–1.20 (m, 4H, 2 × CH_2_), 1.18 (d, *J* = 3.7 Hz, 1H, CH), 1.12 (s, 3H, CH_3_), 1.07–1.01 (m, 1H, CH), 0.95–0.90 (m, 6H, 2 × CH_3_), 0.85 (s, 3H, CH_3_), 0.79 (s, 3H, CH_3_), 0.64 (s, 3H, CH_3_), 0.60 (s, 3H, CH_3_); **^13^C NMR (75 MHz, CDCl_3_)**δ_C_ 177.57 (C=O), 167.71 (C=O), 165.88 (2 × NC=O), 143.88 (C_q_), 136.57 (2 × C_q_), 134.23 (2 × CH_aro_), 132.19 (2 × C_q_), 128.55 (2 × CH_aro_), 128.10 (2 × CH_aro_), 128.04 (CH_aro_), 124.55 (CH_aro/Trz_), 123.59 (2 × CH_aro_), 122.41 (CH=C), 83.97 (CHCO_2_), 66.05 (OCH_2_Ph), 55.27 (CH), 51.31 (TrzCH_2_CO_2_), 47.61 (CH), 46.87 (C_q_), 45.98 (CH_2_), 41.80 (C_q_), 41.50 (C=C**C**HCH_2_), 39.39 (C_q_), 38.07 (CH_2_), 37.79 (C_q_), 36.95 (C_q_), 33.98 (NCH_2_Trz), 33.23 (CH_3_), 33.09 (CH_2_), 32.69 (CH_2_), 32.49 (CH_2_), 30.83 (C_q_), 28.15 (CH_3_), 27.73 (CH_2_), 25.97 (CH_3_), 23.78 (CH_3_), 23.49 (2 × CH_2_), 23.16 (CH_2_), 18.22 (CH_2_), 16.98 (CH_3_), 16.54 (CH_3_), 15.41 (CH_3_); HRMS (+ESI) calculated for C_50_H_63_N_4_O_6_ [M + H]^+^: 815.4703, found: 815.4775.

*(4aS,6aS,6bR,8aR,10S,12aR,12bR,14bS)-benzyl 2,2,6a,6b,9,9,12a-heptamethyl-10-(2-(4-((3,4,4-trimethyl-2,5-dioxoimidazolidin-1-yl)methyl)-1H-1,2,3-triazol-1-yl)acetoxy)-1,2,3,4,4a,5,6,6a,6b,7,8,8a,9,10,11,12,12a,12b,13,14b-icosahydropicene-4a-carboxylate* (**18b**), This derivative was isolated as a white solid in 92% yield; Rf (cyclohexane/EtOAc: 60/40) = 0.4; mp = 92–94 °C; [α]_D_^20^ +51 (c 1 mg/mL, CH_2_Cl_2_); **IR (ν_max_/cm^−1^)** 2923.14 (CH str.), 1770.86 (2 × C=O), 1711.07 (2 × C=O); **^1^H NMR (300 MHz, CDCl_3_)**δ_H_ 7.72 (s, 1H, CH_aro/Trz_), 7.38–7.28 (m, 5H, 5 × CH_aro_), 5.27 (t, *J* = 3.5 Hz, 1H, CH=C), 5.17–5.00 (m, 4H, TrzCH_2_CO_2_, OCH_2_Ph), 4.81 (s, 2H, NCH_2_Trz), 4.55 (dd, *J* = 9.9, 6.0 Hz, 1H, CHCO_2_), 2.94–2.83 (m, 4H, C=CC**H**CH_2_,NCH_3_), 2.02–1.78 (m, 4H, 2 × CH_2_),1.72–1.65 (m, 2H, CH_2_), 1.55–1.42 (m, 4H, 2 × CH_2_), 1.36 (s, 6H, 2 × COCH_3_CN), 1.30–1.19 (m, 10H, 5 × CH_2_), 1.16 (d, *J* = 3.7 Hz, 1H, CH), 1.10 (s, 3H, CH_3_), 1.04–1.00 (m, 1H, CH), 0.93–0.78 (m, 12H, 4 × CH_3_), 0.71 (s, 3H, CH_3_), 0.58 (s, 3H, CH_3_); **^13^C NMR (75 MHz,CDCl_3_)**δ_C_ 177.45 (C=O), 176.03 (NC=O), 165.81 (C=O), 154.68 (NC=O), 143.75 (C_q_), 143.01 (C_q_), 136.43 (C_q_), 128.42 (2 × CH_aro_), 127.98 (2 × CH_aro_), 127.92 (CH_aro_), 124.38 (CH_aro/Trz_), 122.29 (CH=C), 83.79 (CHCO_2_), 65.93 (OCH_2_Ph), 61.39 (C_q_), 55.17 (CH), 51.12 (TrzCH_2_CO_2_), 47.49 (CH), 46.73 (C_q_), 45.84 (CH_2_), 41.68 (C^q^), 41.36 (C=C**C**HCH_2_), 39.27 (C^q^), 37.97 (CH_2_), 37.73 (C_q_), 36.85 (C_q_), 33.80 (NCH_2_Trz), 33.11 (CH_3_), 32.57 (CH_2_), 32.36 (CH_2_), 30.71 (C^q^), 29.71 (CH_2_), 28.09 (CH_3_), 27.59 (CH_2_), 25.85 (CH_3_), 24.41 (NCH_3_), 23.65 (CH_3_), 23.38 (2 × CH_2_), 23.03 (CH_2_), 22.00 (2 × CO**C**H_3_CN), 18.14 (CH_2_), 16.85 (CH_3_), 16.56 (CH_3_), 15.31 (CH_3_); HRMS (+ESI) calculated for C_48_H_68_N_5_O_6_ [M + H]^+^: 810.5125, found: 810.5169.

*(4aS,6aS,6bR,8aR,10S,12aR,12bR,14bS)-benzyl 10-(2-(4-((2,5-dioxopyrrolidin-1-yl)methyl)-1H-1,2,3-triazol-1-yl)acetoxy)-2,2,6a,6b,9,9,12a-heptamethyl-1,2,3,4,4a,5,6,6a,6b,7,8,8a,9,10,11,12,12a,12b,13,14b-icosahydropicene-4a-carboxylate* (**18c**), This derivative was isolated as a white solid in 83% yield; Rf (cyclohexane/EtOAc: 50/50) = 0.3; mp = 145–147 °C; [α]_D_^20^+108 (c 0.5 mg/mL, CH_2_Cl_2_); **IR (ν_max_/cm^−1^)** 2926.82 (CH str.), 1741.39 (C=O), 1721.47 (C=O), 1701.29 (2 × C=O); **^1^H NMR (300 MHz, CDCl_3_)** δ_H_ 7.71 (s, 1H, CH_aro/Trz_), 7.38–7.28 (m, 5H, 5 × CH_aro_), 5.27 (t, *J* = 3.6 Hz, 1H, CH=C), 5.17–4.99 (m, 4H, OCH_2_Ph, NCH_2_Trz), 4.80 (s, 2H, TrzCH_2_CO_2_), 4.54 (dd, *J* = 9.8, 6.1 Hz, 1H, CHCO_2_), 2.88 (dd, *J* = 13.6, 4.5 Hz, 1H, C=CC**H**CH_2_), 2.71 (s, 4H, 2 × CH_2_CON), 2.03–1.79 (m, 4H, 2 × CH_2_), 1.72–1.62 (m, 4H, 2 × CH_2_), 1.56–1.41 (m, 4H, 2 × CH_2_), 1.38–1.18 (m, 8H, 4 × CH_2_), 1.15 (d, *J* = 3.9 Hz, 1H, CH), 1.10 (s, 3H, CH_3_), 1.04–0.99 (m, 1H, CH), 0.93–0.85 (m, 9H, 3 × CH_3_), 0.81 (s, 3H, CH_3_), 0.71 (s, 3H, CH_3_), 0.58 (s, 3H, CH_3_); **^13^C NMR (75 MHz,CDCl_3_)**δ_C_ 177.51 (C=O), 176.54 (2 × NC=O), 165.90 (C=O), 143.82 (C_q_), 142.37 (C_q_), 136.49 (C_q_), 128.49 (2 × CH_aro_), 128.06 (2 × CH_aro_), 127.99 (CH_aro_), 124.72 (CH_aro/Trz_), 122.35 (CH=C), 83.91 (CHCO_2_), 66.00 (OCH_2_Ph), 55.24 (CH), 51.16 (TrzCH_2_CO_2_), 47.56 (CH), 46.79 (C_q_), 45.91 (CH_2_), 41.75 (C_q_), 41.43 (C=C**C**HCH_2_), 39.33 (C_q_), 38.03 (CH_2_), 37.80 (C_q_), 36.92 (C_q_), 33.92 (NCH_2_Trz), 33.68 (CH_2_), 33.18 (CH_3_), 32.63 (CH_2_), 32.43 (CH_2_), 30.78 (C^q^), 28.30 (2 × **C**H_2_CON), 28.16 (CH_3_), 27.66 (CH_2_), 25.93 (CH_3_), 23.73 (CH_3_), 23.46 (2 × CH_2_), 23.10 (CH_2_), 18.22 (CH_2_), 16.92 (CH_3_), 16.60 (CH_3_), 15.40 (CH_3_); HRMS (+ESI) calculated for C_46_H_63_N_4_O_6_ [M + H]^+^: 767.4703, found: 767.4751.

*(4aS,6aS,6bR,10S,12aR)-benzyl 10-(2-(4-((1,1-dioxido-3-oxobenzo[d]isothiazol-2(3H)-yl)methyl)-1H-1,2,3-triazol-1-yl)acetoxy)-2,2,6a,6b,9,9,12a-heptamethyl-1,2,3,4,4a,5,6,6a,6b,7,8,8a,9,10,11,12,12a,12b,13,14b-icosahydropicene-4a-carboxylate* (**18d**), This derivative was isolated as a white solid in 89% yield; Rf (cyclohexane/EtOAc: 60/40) = 0.5; mp = 113–116 °C; [α]_D_^20^ +80 (c 0.7 mg/mL, CH_2_Cl_2_); **IR (ν_max_/cm^−1^)** 2921.46 (CH str.), 1725.07 (3 × C=O), 1181.72 (2 × S=O); **^1^H NMR (300 MHz, CDCl_3_**)δ_H_ 8.12–7.78 (m, 5H_,_ 5 × CH_aro_), 7.39–7.28 (m, 5H, 5 × CH_aro_), 5.27 (t, *J* = 3.69 Hz, 1H, CH=C), 5.21–4.97 (m, 6H, NCH_2_Trz, TrzCH_2_CO_2_, OCH_2_Ph), 4.53 (dd, *J* = 10.2, 5.6 Hz, 1H, CHCO_2_), 2.89 (dd, *J* = 13.7, 4.4 Hz, 1H, C=CC**H**CH_2_), 2.03–1.77 (m, 4H, 2 × CH_2_), 1.76–1.63 (m, 4H, 2 × CH_2_), 1.59–1.48 (m, 4H, 2 × CH_2_) 1.47–1.19 (m, 8H, 4 × CH_2_), 1.16 (d, *J* = 3.3 Hz, 1H, CH), 1.10 (s, 3H, CH_3_), 1.01 (m, 1H, CH), 0.93–0.87 (m, 6H, 2 × CH_3_), 0.83 (s, 3H, CH_3_), 0.78 (s, 3H, CH_3_), 0.65 (s, 3H, CH_3_), 0.58 (s, 3H, CH_3_); **^13^C NMR (75 MHz,CDCl_3_)**δ_C_ 177.56 (C=O), 165.77 (C=O), 158.59 (NC=O), 143.87 (C_q_), 137.87 (C_q_), 136.56 (2 × C_q_), 135.09 (CH_aro_), 134.56 (CH_aro_), 128.54 (2 × CH_aro_), 128.09 (2 × CH_aro_), 128.03 (CH_aro_), 127.31 (C_q_), 125.48 (2 × CH_aro_), 122.40 (CH_aro/Trz_), 121.24 (CH=C), 84.00 (CHCO_2_), 66.04 (OCH_2_Ph), 55.28 (CH), 47.59 (CH), 46.85 (C_q_), 45.97 (TrzCH_2_CO_2_), 45.94 (CH_2_), 41.79 (C_q_), 41.49 (C=C**C**HCH_2_), 39.38 (C_q_), 38.07 (CH_2_), 37.80 (C_q_), 36.94 (C_q_), 33.98 (NCH_2_Trz), 33.96 (CH_2_), 33.21 (CH_3_), 32.67 (CH_2_), 32.47 (CH_2_), 30.82 (C_q_), 28.18 (CH_3_), 27.72 (CH_2_), 25.96 (CH_3_), 23.76 (CH_3_), 23.48 (2 × CH_2_), 23.15 (CH_2_), 18.22 (CH_2_), 16.97 (CH_3_), 16.59 (CH_3_), 15.38 (CH_3_); HRMS (+ESI) calculated for C_49_H_63_N_4_O_7_S [M + H]^+^: 851.4373, found: 851.4464.

*(4aS,6aS,6bR,8aR,10S,12aR,12bR,14bS)-benzyl 10-(2-(4-(2-(1,3-dioxoisoindolin-2-yl)ethyl)-1H-1,2,3-triazol-1-yl)acetoxy)-2,2,6a,6b,9,9,12a-heptamethyl-1,2,3,4,4a,5,6,6a,6b,7,8,8a,9,10,11,12,12a,12b,13,14b-icosahydropicene-4a-carboxylate* (**18e**), This derivative was isolated as a white solid in 92% yield; Rf (cyclohexane/EtOAc: 60/40) = 0.4; mp = 94–96 °C; [α]_D_^20^ +65 (c 0.85 mg/mL, CH_2_Cl_2_); **IR (ν_max_/cm^−1^)** 2929.05 (CH str.), 1714.31 (4 × C=O); **^1^H NMR (300 MHz, CDCl_3_)**δ_H_ 7.86–7.79 (m, 2H, 2 × CH_aro_), 7.74–7.67 (m, 2H, 2 × CH_aro_), 7.56 (s, 1H, 1H, CH_aro/Trz_), 7.39–7.28 (m, 5H, 5 × CH_aro_), 5.27 (t, *J* = 3.6 Hz, 1H, CH=C), 5.14–4.99 (m, 4H, TrzCH_2_CO_2_, OCH_2_Ph), 4.54 (dd, *J* = 10.0, 5.9 Hz, 1H, CHCO_2_), 4.02 (t, *J* = 7.3 Hz, 2H, NC**H_2_**CH_2_Trz), 3.17 (t, *J* = 7.3 Hz, 2H, NCH_2_C**H_2_**Trz), 2.89 (dd, *J* = 14.0, 4.5 Hz, 1H, C=CC**H**CH_2_), 2.02–1.76 (m, 4H, 2 × CH_2_), 1.75–1.66 (m, 2H, CH_2_), 1.60–1.55 (m, 2H, CH_2_) 1.55–1.39 (m, 4H, 2 × CH_2_), 1.38–1.19 (m, 8H, 4 × CH_2_), 1.16 (d, *J* = 4.0 Hz, 1H, CH), 1.11 (s, 3H, CH_3_), 1.04–0.99 (m, 1H, CH_3_), 0.92–0.86 (m, 9H, 3 × CH_3_), 0.81 (s, 3H, CH_3_), 0.71 (s, 3H, CH_3_), 0.58 (s, 3H, CH_3_); **^13^C NMR (75 MHz,CDCl_3_)**δ_C_ 177.55 (C=O), 168.28 (C=O), 166.13 (2 × NC=O), 144.80 (C_q_), 143.86 (C_q_), 136.54 (C_q_), 134.08 (2 × CH_aro_), 132.17 (2 × C_q_), 128.53 (2 × CH_aro_), 128.09 (2 × CH_aro_), 128.03 (CH_aro_), 123.43 (2 × CH_aro_), 122.88 (CH_aro/Trz_), 122.40 (CH=C), 83.84 (CHCO_2_), 66.04 (OCH_2_Ph), 55.30 (CH), 51.20 (TrzCH_2_CO_2_), 47.60 (CH), 46.84 (C_q_), 45.96 (CH_2_), 41.79 (C_q_), 41.48 (C=C**C**HCH_2_), 39.38 (C_q_), 38.08 (CH_2_), 37.84 (C_q_), 37.45 (N**C**H_2_CH_2_Trz), 36.96 (C_q_), 33.96 (CH_2_), 33.21 (CH_3_), 32.68 (CH_2_), 32.47 (CH_2_), 30.81 (C_q_), 28.19 (CH_3_), 27.71 (CH_2_), 25.96 (CH_3_), 24.96 (NCH_2_**C**H_2_Trz), 23.76 (CH_3_), 23.49 (2 × CH_2_), 23.14 (CH_2_), 18.25 (CH_2_), 16.96 (CH_3_), 16.64 (CH_3_), 15.43 (CH_3_); HRMS (+ESI) calculated for C_51_H_65_N_4_O_6_ [M + H]^+^: 829.4859, found: 829.4959.

*(4aS,6aS,6bR,8aR,10S,12aR,12bR,14bS)-benzyl 10-(2-(4-(1-(1,3-dioxoisoindolin-2-yl)ethyl)-1H-1,2,3-triazol-1-yl)acetoxy)-2,2,6a,6b,9,9,12a-heptamethyl-1,2,3,4,4a,5,6,6a,6b,7,8,8a,9,10,11,12,12a,12b,13,14b-icosahydropicene-4a-carboxylate* (**18f**), This derivative was isolated as a white solid in 80% yield; Rf (cyclohexane/EtOAc: 60/40) = 0.5; mp = 82–84 °C; [α]_D_^20^ +40 (*c* 1 mg/mL, CH_2_Cl_2_); **IR (ν_max_/cm^−1^)**2942.80 (CH str.), 1776.94 (C=O), 17113.95 (3 × C=O); **^1^H NMR (300 MHz, CDCl_3_)** δ_H_ 7.84–7.77 (m, 3H, 3 × CH_aro_), 7.73–7.66 (m, 2H, 2 × CH_aro_), 7.38–7.28 (m, 5H, 5 × CH_aro_), 5.81 (q, *J* = 7.3 Hz, 1H, NC**H**CH_3_), 5.27 (t, *J* = 3.6 Hz, 1H, CH=C), 5.21–5.00 (m, 4H, TrzCH_2_CO_2_, OCH_2_Ph), 4.53 (dd, *J* = 10.5, 5.4 Hz, 1H, CHCO_2_), 2.89 (dd, *J* = 13.6, 4.4 Hz, 1H, C=CC**H**CH_2_), 2.06–1.89 (m, 2H, CH_2_), 1.86 (d, *J* = 7.3 Hz, 3H, NCHC**H_3_**), 1.84–1.76 (m, 2H, CH_2_), 1.74–1.63 (m, 4H, 2 × CH_2_) 1.58–1.44 (m, 4H, 2 × CH_2_), 1.42–1.18 (m, 8H, 4 × CH_2_),1.16 (d, *J* = 3.7 Hz, 1H, CH), 1.10 (s, 3H, CH_3_), 1.01 (d, *J* = 10.7 Hz, 1H, CH), 0.93–0.87 (m, 6H, 2 × CH_3_), 0.84 (s, 3H, CH_3_), 0.78 (s, 3H, CH_3_), 0.64 (s, 3H, CH_3_), 0.58 (s, 3H, CH_3_); **^13^C NMR (75 MHz,CDCl_3_)** δ_C_ 177.54 (C=O), 167.76 (C=O), 165.67 (2 × NC=O), 147.82 (C_q_), 143.85 (C_q_), 136.53 (2 × C_q_), 134.13 (2 × CH_aro_), 132.05 (C_q_), 128.53 (2 × CH_aro_), 128.08 (2 × CH_aro_), 128.02 (CH_aro_), 123.84 (CH_aro/Trz_), 123.40 (2 × CH_aro_), 122.39 (CH=C), 83.94 (CHCO_2_), 66.03 (OCH_2_Ph), 55.26 (CH), 51.32 (TrzCH_2_CO_2_), 47.58 (CH), 46.83 (C_q_), 45.94 (CH_2_), 42.62 (N**C**HCH_3_), 41.77 (C_q_), 41.46 (C=C**C**HCH_2_), 39.35 (C_q_), 38.05 (CH_2_), 37.78 (C_q_), 36.93 (C_q_), 33.95 (CH_2_), 33.21 (CH_3_), 32.65 (CH_2_), 32.45 (CH_2_), 30.81 (C_q_), 28.15 (CH_3_), 27.70 (CH_2_), 25.95 (CH_3_), 23.75 (CH_3_), 23.46 (2 × CH_2_), 23.13 (CH_2_), 18.42 (NCH**C**H_3_), 18.21 (CH_2_), 16.95 (CH_3_), 16.58 (CH_3_), 15.40 (CH_3_); HRMS (+ESI) calculated for C_51_H_65_N_4_O_6_ [M + H]^+^: 829.4859, found: 829.4942.

*(4aS,6aS,6bR,8aR,10S,12aR,12bR,14bS)-benzyl 10-(2-(4-(((1,3-dioxoisoindolin-2-yl)oxy)methyl)-1H-1,2,3-triazol-1-yl)acetoxy)-2,2,6a,6b,9,9,12a-heptamethyl-1,2,3,4,4a,5,6,6a,6b,7,8,8a,9,10,11,12,12a,12b,13,14b-icosahydropicene-4a-carboxylate* (**18g**), This derivative was isolated as a white solid in 89% yield; Rf (cyclohexane/EtOAc: 60/40) = 0.4; mp = 107–109 °C; [α]_D_^20^ +77 (*c* 0.7 mg/mL, CH_2_Cl_2_); **IR (ν_max_/cm^−1^)** 2945.81 (CH str.), 1791.82 (C=O), 1729.79 (3 × C=O); **^1^H NMR (300 MHz, CDCl_3_)**δ_H_ 8.00 (s, 1H, CH_aro/Trz_), 7.82–7.69 (m, 4H, 4 × CH_aro_), 7.38–7.28 (m, 5H, 5 × CH_aro_), 5.39 (s, 2H, OCH_2_Trz), 5.28 (t, *J* = 3.5 Hz, 1H, CH=C), 5.17 (s, 2H, OCH_2_Ph), 5.12–4.99 (m, 2H, TrzCH_2_CO_2_), 4.58 (t, *J* = 7.9 Hz, 1H, CHCO_2_), 2.89 (dd, *J* = 13.9, 4.3 Hz, 1H, C=CC**H**CH_2_), 2.09–1.73 (m, 4H, 2 × CH_2_) 1.71–1.61 (m, 6H, 3 × CH_2_), 1.52–1.43 (m, 2H, CH_2_), 1.42–1.18 (m, 8H, 4 × CH_2_), 1.16 (d, *J* = 3.7 Hz, 1H, CH), 1.11 (s, 3H, CH_3_), 1.05–1.00 (m, 1H, CH), 0.94–0.82 (m, 12H, 4 × CH_3_), 0.75 (s, 3H, CH_3_), 0.59 (s, 3H, CH_3_); **^13^C NMR (75 MHz, CDCl_3_)**δ_C_ 177.56 (C=O), 165.87 (C=O), 163.56 (2 × NC=O), 143.86 (C_q_), 142.13 (C_q_), 136.54 (C_q_), 134.61 (2 × CH_aro_), 128.95 (2 × C_q_), 128.53 (2 × CH_aro_), 128.09 (2 × CH_aro_), 128.03 (CH_aro_), 126.09 (CH_aro/Trz_), 123.73 (2 × CH_aro_), 122.39 (CH=C), 84.05 (CHCO_2_), 70.52 (OCH_2_Trz), 66.04 (OCH_2_Ph), 55.30 (CH), 51.30 (TrzCH_2_CO_2_), 47.60 (CH), 46.84 (C_q_), 45.95 (CH_2_), 41.79 (C_q_), 41.47 (C=C**C**HCH_2_), 39.37 (C_q_), 38.09 (CH_2_), 37.87 (C_q_), 36.96 (C_q_), 33.96 (CH_2_), 33.21 (CH_3_), 32.67 (CH_2_), 32.46 (CH_2_), 30.82 (C_q_), 28.22 (CH_3_), 27.70 (CH_2_), 25.96 (CH_3_), 23.76 (CH_3_), 23.51 (2 × CH_2_), 23.14 (CH_2_), 18.25 (CH_2_), 16.96 (CH_3_), 16.68 (CH_3_), 15.44 (CH_3_); HRMS (+ESI) calculated for C_50_H_63_N_4_O_7_ [M + H]^+^: 831.4652, found: 831.4703.

*(4aS,6aS,6bR,8aR,10S,12aR,12bR,14bS)-benzyl 10-(2-(4-((1-hydroxy-3-oxoisoindolin-2-yl)methyl)-1H-1,2,3-triazol-1-yl)acetoxy)-2,2,6a,6b,9,9,12a-heptamethyl-1,2,3,4,4a,5,6,6a,6b,7,8,8a,9,10,11,12,12a,12b,13,14b-icosahydropicene-4a-carboxylate* (**18h**), This derivative was isolated as a white solid in 96% yield; Rf (cyclohexane/EtOAc: 60/40) = 0.4; mp = 88–90 °C; [α]_D_^20^ +24 (*c* 1 mg/mL, CH_2_Cl_2_); **IR (ν_max_/cm^−1^)** 3350 (OH), 2928.90 (CH str.), 1699.10 (3 × C=O); **^1^H NMR (300 MHz, CDCl_3_)**δ_H_7.77 (s, 1H, CH_aro/Trz_), 7.70 (d, *J* = 7.4 Hz, 1H,CH_aro_), 7.61–7.40 (m, 3H, 3 × CH_aro_), 7.39–7.27 (m, 5H, 5 × CH_aro_), 5.92 (s, 1H, C**H**OH), 5.25 (t, *J* = 3.8 Hz, 1H, CH=C), 5.17–4.98 (m, 4H, TrzCH_2_CO_2_,OCH_2_Ph), 4.90 (d, *J* = 13.4 Hz, 1H, NCH_2_^b^Trz), 4.70 (d, *J* = 15.4 Hz, 1H, NCH_2_^a^Trz), 4.49 (dd, *J* = 10.5, 5.5 Hz, 1H, CHCO_2_), 2.89 (dd, *J* = 14.0, 4.4 Hz, 1H, C=CC**H**CH_2_), 2.19–1.77 (m, 4H, 2 × CH_2_), 1.76–1.61 (m, 4H, 2 × CH_2_), 1.53–1.38 (m, 4H, 2 × CH_2_), 1.37–1.19 (m, 8H, 4 × CH_2_), 1.16 (d, *J* = 4.4 Hz, 1H, CH), 1.09 (s, 3H, CH_3_), 1.03–0.99 (m, 1H, CH), 0.92–0.86 (m, 6H, 2 × CH_3_), 0.79 (s, 3H, CH_3_), 0.76 (s, 3H, CH_3_), 0.62 (s, 3H, CH_3_), 0.57 (s, 3H, CH_3_); **^13^C NMR (75 MHz,CDCl_3_)** δ_C_ 177.53 (C=O), 167.40 (NC=O), 165.89 (C=O), 144.25 (C_q_), 143.80 (C_q_), 136.49 (C_q_), 132.42 (C_q_), 131.46 (C_q_), 129.65 (2 × CH_aro_), 128.50 (2 × CH_aro_), 128.04 (2 × CH_aro_), 127.99 (CH_aro_), 123.60 (2 × CH_aro_), 123.26 (CH_aro/Trz_), 122.35 (CH=C), 83.93 (CHCO_2_), 81.59 (CHOH), 66.00 (OCH_2_Ph), 55.19 (CH), 51.25 (TrzCH_2_CO_2_), 47.53 (CH), 46.79 (C_q_), 45.91 (CH_2_), 41.73 (C_q_), 41.42 (C=C**C**HCH_2_), 39.32 (C_q_), 37.99 (CH_2_), 37.73 (C_q_), 36.87 (C_q_), 34.49 (NCH_2_Trz), 33.92 (CH_2_), 33.18 (CH_3_), 32.61 (CH_2_), 32.42 (CH_2_), 30.77 (C_q_), 28.09 (CH_3_), 27.66 (CH_2_), 25.93 (CH_3_), 23.73 (CH_3_), 23.42 (2 × CH_2_), 23.10 (CH_2_), 18.15 (CH_2_), 16.91 (CH_3_), 16.51 (CH_3_), 15.35 (CH_3_); HRMS (+ESI) calculated for C_50_H_64_N_4_O_6_Na [M+Na]^+^: 839.4718, found: 839.4752.

*(4aS,6aS,6bR,8aR,10S,12aR,12bR,14bS)-benzyl 10-(2-(4-(2-(1-hydroxy-3-oxoisoindolin-2-yl)ethyl)-1H-1,2,3-triazol-1-yl)acetoxy)-2,2,6a,6b,9,9,12a-heptamethyl-1,2,3,4,4a,5,6,6a,6b,7,8,8a,9,10,11,12,12a,12b,13,14b-icosahydropicene-4a-carboxylate* (**18i**), This derivative was isolated as a white solid in 98% yield; Rf (cyclohexane/EtOAc: 40/60) = 0.5; mp = 111–113 °C; [α]_D_^20^ +40 (*c* 1 mg/mL, CH_2_Cl_2_); **IR (ν_max_/cm^−1^)**3350 (OH), 2931.46 (CH str.), 1729.50 (3 × C=O); **^1^H NMR (300 MHz, CDCl_3_)** δ_H_ 7.68 (d, *J* = 7.4 Hz, 1H, CH_aro/Trz_), 7.59–7.40 (m, 4H, 4 × CH_aro_), 7.37–7.28 (m, 5H, 5 × CH_aro_), 5.83 (d, *J* = 6.6 Hz, 1H, C**H**OH), 5.39 (br. s, 1H, OH), 5.27 (t, *J* = 3.7 Hz, CH=C), 5.14–4.98 (m, 4H, TrzCH_2_CO_2_,OCH_2_Ph), 4.52 (t, *J* = 8.0 Hz, 1H, CHCO_2_), 3.92 (t, *J* = 6.7 Hz, 2H, NC**H_2_**CH_2_Trz), 3.17 (t, *J* = 6.8 Hz, 2H, NCH_2_C**H_2_**Trz), 2.89 (dd, *J* = 13.8, 4.5 Hz, 1H, C=CC**H**CH_2_), 2.04–1.79 (m, 4H, 2 × CH_2_), 1.74–1.67 (m, 2H, CH_2_), 1.53–1.47 (m, 2H, CH_2_), 1.47–1.35 (m, 4H, 2 × CH_2_), 1.31–1.21 (m, 8H, 4 × CH_2_),1.16 (d, *J* = 3.8 Hz, 1H, CH), 1.11 (s, 3H, CH_3_), 1.04–1.00 (m, 1H, CH), 0.92–0.85 (m, 9H, 3 × CH_3_), 0.79 (s, 3H, CH_3_), 0.72 (s, 3H, CH_3_), 0.58 (s, 3H, CH_3_); **^13^C NMR (75 MHz,CDCl_3_)** δ_C_ 177.57 (C=O), 167.70 (C=O), 166.06 (NC=O), 144.23 (C_q_), 143.87 (C_q_), 136.52 (C_q_), 132.26 (CH_aro_), 131.67 (2 × C_q_), 129.63 (CH_aro_), 128.53 (2 × CH_aro_), 128.09 (2 × CH_aro_), 128.03 (CH_aro_), 123.48 (2 × CH_aro_), 123.20 (CH_aro/Trz_), 122.38 (CH=C), 84.10 (CHCO_2_), 82.51 (C**H**OH), 66.04 (OCH_2_Ph), 55.27 (CH), 51.39 (TrzCH_2_CO_2_), 47.59 (CH), 46.83 (C_q_), 45.95 (CH_2_), 41.78 (C_q_), 41.46 (C=C**C**HCH_2_), 39.36 (C_q_), 38.93 (N**C**H_2_CH_2_Trz), 38.06 (CH_2_), 37.84 (C_q_), 36.94 (C_q_), 33.95 (CH_2_), 33.22 (CH_3_), 32.65 (CH_2_), 32.46 (CH_2_), 30.82 (C_q_), 28.20 (CH_3_), 27.70 (CH_2_), 25.97 (CH_3_), 24.87 (NCH_2_**C**H_2_Trz), 23.76 (CH_3_), 23.49 (2 × CH_2_), 23.13 (CH_2_), 18.24 (CH_2_), 16.95 (CH_3_), 16.65 (CH_3_), 15.44 (CH_3_); HRMS (+ESI) calculated for C_51_H_67_N_4_O_6_ [M + H]^+^: 831.5016, found: 831.5037.

*Methyl2-((1-(2-(((3S,4aR,6aR,6bS,8aS,12aS,14aR,14bR)-8a-((benzyloxy)carbonyl)-4,4,6a,6b,11,11,14b-heptamethyl-1,2,3,4,4a,5,6,6a,6b,7,8,8a,9,10,11,12,12a,14,14a,14b-icosahydropicen-3-yl)oxy)-2-oxoethyl)-1H-1,2,3-triazol-4-yl)methoxy)-3-oxoisoindoline-1-carboxylate* (**18j**), This derivative was isolated as a white solid in 95% yield; Rf (cyclohexane/EtOAc: 50/50) = 0.5; mp = 86–88 °C; [α]_D_^20^ +52 (*c* 0.9 mg/mL, CH_2_Cl_2_); **IR (ν_max_/cm^−1^)** 2926.94 (CH str.), 1732.76 (4 × C=O); **^1^H NMR (300 MHz, CDCl_3_)** δ_H_ 7.96 (s, 1H, CH_aro/Trz_), 7.79 (d, *J* = 7.8 Hz, 1H, CH_aro_), 7.64–7.43 (m, 3H, 3 × CH_aro_), 7.38–7.30 (m, 5H, 5 × CH_aro_), 7.03 (s, 1H, CH_3_CO_2_C**H**), 5.33 (d, *J* = 1.7 Hz, 2H, OCH_2_Trz), 5.28 (t, *J* = 4.0 Hz, 1H, CH=C), 5.17 (s, 2H, OCH_2_Ph), 5.13–4.99 (m, 2H, TrzCH_2_CO_2_), 4.57 (dd, *J* = 9.4, 6.6 Hz, 1H, CHCO_2_), 2.90 (dd, *J* = 14.0, 4.4 Hz, 1H, C=CC**H**CH_2_), 2.17 (s, 3H, CH_3_CO_2_), 1.51–1.38 (m, 4H, 2 × CH_2_), 1.32–1.23 (m, 8H, 4 × CH_2_), 1.16 (d, *J* = 3.7 Hz, 1H, CH), 1.11 (s, 3H, CH_3_), 1.04–0.99 (m, 1H, CH), 0.93–0.85 (m, 9H, 3 × CH_3_), 0.83 (s, 3H, CH_3_), 0.73 (s, 3H, CH_3_), 0.59 (s, 3H, CH_3_); **^13^C NMR (75 MHz,CDCl_3_)** δ_C_ 177.56 (C=O), 170.81 (C=O), 165.92 (C=O), 165.66 (NC=O), 143.88 (C_q_), 142.91 (C_q_), 138.67 (C_q_), 136.56 (C_q_), 133.51 (CH_aro_), 130.71 (CH_aro_), 129.34, (C_q_), 128.54 (2 × CH_aro_), 128.10 (2 × CH_aro_), 128.04 (CH_aro_), 125.85 (CH_aro_), 124.22 (CH_aro_), 124.06 (CH_aro/Trz_), 122.41 (CH=C), 84.01 (CHCO_2_), 80.82 (CH_3_CO_2_**C**H), 66.05 (OCH_2_Ph), 62.01 (OCH_2_Trz), 55.31 (CH), 51.27 (TrzCH_2_CO_2_), 47.61 (CH), 46.85 (C_q_), 45.96 (CH_2_), 41.80 (C_q_), 41.49 (C=C**C**HCH_2_), 39.39 (C_q_), 38.09 (CH_2_), 37.87 (C_q_), 36.97 (C_q_), 33.98 (CH_2_), 33.22 (CH_3_), 32.69 (CH_2_), 32.48 (CH_2_), 30.83 (C_q_), 28.22 (CH_3_), 27.72 (CH_2_), 25.97 (CH_3_), 23.77 (CH_3_), 23.52 (2 × CH_2_), 23.15 (CH_2_), 21.18 (CH_3_CO_2_), 18.26 (CH_2_), 16.97 (CH_3_), 16.66 (CH_3_), 15.45 (CH_3_); HRMS (+ESI) calculated for C_52_H_67_N_4_O_8_ [M + H]^+^: 875.4914, found: 875.4983.

*Ethyl2-((1-(2-(((3S,4aR,6aR,6bS,8aS,12aS,14aR,14bR)-8a-((benzyloxy)carbonyl)-4,4,6a,6b,11,11,14b-heptamethyl-1,2,3,4,4a,5,6,6a,6b,7,8,8a,9,10,11,12,12a,14,14a,14b-icosahydropicen-3-yl)oxy)-2-oxoethyl)-1H-1,2,3-triazol-4-yl)methyl)-3-oxoisoindoline-1-carboxylate* (**18k**), This derivative was isolated as a white solid in 71% yield; Rf (cyclohexane/EtOAc: 60/40) = 0.5; mp = 91–93 °C; [α]_D_^20^ +47 (*c* 0.95 mg/mL, CH_2_Cl_2_); **IR (ν_max_/cm^−1^)** 2943.22 (CH str.), 1702.18 (4 × C=O); **^1^H NMR (300 MHz, CDCl_3_)**δ_H_ 7.82 (d, *J* = 7.3 Hz, 1H, CH_aro_), 7.74 (s, 1H, CH_aro/Trz_), 7.65–7.45 (m, 3H, 3 × CH_aro_), 7.39–7.29 (m, 5H, 5 × CH_aro_), 5.43–5.34 (m, 1H, OCH_2_^b^Ph), 5.31–5.23 (m, 2H, CH=C, OCH_2_^a^Ph), 5.14–5.00 (m, 4H, NCH_2_^b^Trz, C**H**CO_2_CH_2_CH_3_, TrzCH_2_CO_2_), 4.62–4.46 (m, 2H, CHCO_2_, NCH_2_^a^Trz), 4.40–4.20 (m, 2H, CH_3_C**H_2_**O), 2.91 (dd, *J* = 13.2, 4,1 Hz, 1H, C=CC**H**CH_2_), 2.07–1.72 (m, 4H, 2 × CH_2_), 1.70–1.55 (m, 8H, 4 × CH_2_), 1.54–1.37 (m, 4H, 2 × CH_2_), 1.33 (t, *J* = 7.1 Hz, 3H, C**H_3_**CH_2_O), 1.28–1.21 (m, 4H, 2 × CH_2_),1.16 (d, *J* = 3.7 Hz, 1H, CH), 1.09 (s, 3H, CH_3_), 1.05–0.96 (m, 1H, CH), 0.93–0.86 (m, 6H, 2 × CH_3_), 0.82 (s, 3H, CH_3_), 0.74 (s, 3H, CH_3_), 0.61–0.52 (m, 6H, 2 × CH_3_); **^13^C NMR (75 MHz,CDCl_3_)** δ_C_ 177.58 (C=O), 168.50 (C=O), 168.08 (C=O), 165.81 (NC=O), 143.78 (C_q_), 139.72 (C_q_), 135.56 (2 × C_q_), 132.20 (CH_aro_), 129.32 (CH_aro_), 128.92 (C_q_) 128.55 (2 × CH_aro_), 128.09 (2 × CH_aro_), 128.04 (CH_aro_), 124.52 (CH_aro_), 124.05 (CH_aro_), 123.16 (CH_aro/Trz_), 122.40 (CH=C), 83.96 (CHCO_2_), 66.05 (OCH_2_Ph), 62.45 (CH_3_**C**H_2_O), 62.01 (**C**HCO_2_CH_2_CH_3_), 61.15 (NCH_2_Trz), (55.23 (CH), 51.29 (TrzCH_2_CO_2_), 47.58 (CH), 46.85 (C_q_), 45.95 (CH_2_), 41.78 (C_q_), 41.48 (C=C**C**HCH_2_), 39.37 (C_q_), 38.03 (CH_2_), 37.76 (C_q_), 36.92 (C_q_), 33.98 (CH_2_), 33.22 (CH_3_), 32.66 (CH_2_), 32.48 (CH_2_), 30.83 (C_q_), 28.13 (CH_3_), 27.71 (CH_2_), 25.96 (CH_3_), 23.77 (CH_3_), 23.47 (2 × CH_2_), 23.14 (CH_2_), 18.20 (CH_2_), 16.97 (CH_3_), 16.47 (CH_3_), 15.40 (CH_3_), 14.33 (**C**H_3_CH_2_O); HRMS (+ESI) calculated for C_53_H_69_N_4_O_7_ [M + H]^+^: 873.5122, found: 873.5161.

*Ethyl2-benzyl-1-((1-(2-(((3S,4aR,6aR,6bS,8aS,12aS,14aR,14bR)-8a-((benzyloxy)carbonyl)-4,4,6a,6b,11,11,14b-heptamethyl-1,2,3,4,4a,5,6,6a,6b,7,8,8a,9,10,11,12,12a,14,14a,14b-icosahydropicen-3-yl)oxy)-2-oxoethyl)-1H-1,2,3-triazol-4-yl)methyl)-3-oxoisoindoline-1-carboxylate* (**18l**), This derivative was isolated as a white solid in 83% yield; Rf (cyclohexane/EtOAc: 60/40) = 0.5; mp = 105–106 °C; [α]_D_^20^+13 (*c* 0.7 mg/mL, CH_2_Cl_2_); **IR (ν_max_/cm^−1^)** 2927.79 (CH str.), 1710 (2 × C=O), 1700.90 (2 × C=O); **^1^H NMR (300 MHz, CDCl_3_)**δ_H_ 7.78 (d, *J* = 7.5 Hz, 1H_,_ CH_aro_), 7.60–7.39 (m, 5H, 5 × CH_aro_), 7.36–7.27 (m, 6H, 6 × CH_aro_), 7.25–7.20 (m, 1H, CH_aro_), 6.18 (d, *J* = 9.4 Hz, 1H, CH_aro_), 5.27 (t, *J* = 3.7 Hz, 1H, CH=C), 5.13–5.00 (m, 2H, OCH_2_Ph), 4.94–4.62 (m, 4H, 2 × CH_2_, TrzCH_2_CO_2_, NCH_2_Ph), 4.53–4.43 (m, 1H, CHCO_2_), 3.90–3.74 (m, 3H, CH_3_C**H_2_**^b^O, CCH_2_Trz), 3.53 (dqd, *J* = 14.4, 7.1, 5.0 Hz, 1H, CH_3_C**H_2_**^a^O), 2.89 (dd, *J* = 13.6, 4.4 Hz, 1H, C=CC**H**CH_2_), 2.09–1.71 (m, 4H, 2 × CH_2_), 1.70–1.48 (m, 10H, 5 × CH_2_),1.47–1.27 (m, 6H, 3 × CH_2_), 1.25 (t, *J* = 7.5 Hz, 3H, C**H_3_**CH_2_O), 1.18 (d, *J* = 9.3 Hz, 1H, CH), 1.10 (s, 3H, CH_3_), 1.04–0.99 (m, 1H, CH), 0.93–0.87 (m, 9H, 3 × CH_3_), 0.78 (s, 3H, CH_3_), 0.68 (s, 3H, CH_3_), 0.59 (s, 3H, CH_3_); **^13^C NMR (75 MHz,CDCl_3_)** δ_C_177.56 (C=O)_,_ 169.99 (C=O)_,_ 169.38 (C=O), 165.84 (NC=O), 143.87 (C_q_), 143.76 (C_q_), 140.77 (C_q_), 137.17 (C_q_), 136.54 (2 × C_q_), 132.35 (CH_aro_), 129.37 (3 × CH_aro_), 128.53 (4 × CH_aro_), 128.11 (2 × CH_aro_), 128.04 (CH_aro_), 127.69 (CH_aro_), 124.06 (CH_aro_), 123.08 (CH_aro/Trz_), 122.41 (CH=C), 122.00 (CH_aro_), 83.72 (CHCO_2_), 70.76 (C_q_), 66.05 (OCH_2_Ph), 62.33 (CH_3_**C**H_2_O), 55.28 (CH), 50.94 (TrzCH_2_CO_2_), 47.61 (CH), 46.84 (C_q_), 45.96 (CH_2_), 44.87 (NCH_2_Ph), 41.79 (C_q_), 41.47 (C=C**C**HCH_2_), 39.38 (C_q_), 38.08 (CH_2_), 37.86 (C_q_), 36.95 (C_q_), 33.97 (CH_2_), 33.22 (CH_3_), 32.68 (CH_2_), 32.47 (CH_2_), 30.82 (C_q_), 29.85 (C**C**H_2_Trz), 28.18 (CH_3_), 27.71 (CH_2_), 25.96 (CH_3_), 23.76 (CH_3_), 23.48 (2 × CH_2_), 23.15 (CH_2_), 18.26 (CH_2_), 16.96 (CH_3_), 16.65 (CH_3_), 15.43 (CH_3_), 13.61 (**C**H_3_CH_2_O); HRMS (+ESI) calculated for C_60_H_75_N_4_O_7_ [M + H]^+^: 963.5591, found: 963.5641.

*Ethyl1-((1-(2-(((3S,4aR,6aR,6bS,8aS,12aS,14aR,14bR)-8a-((benzyloxy)carbonyl)-4,4,6a,6b,11,11,14b-heptamethyl-1,2,3,4,4a,5,6,6a,6b,7,8,8a,9,10,11,12,12a,14,14a,14b-icosahydropicen-3-yl)oxy)-2-oxoethyl)-1H-1,2,3-triazol-4-yl)methyl)-3-oxo-2-phenethylisoindoline-1-carboxylate* (**18m**), This derivative was isolated as a white solid in 90% yield; Rf (cyclohexane/EtOAc: 70/30) = 0.5; mp = 93–95 °C; [α]_D_^20^ +73 (*c* 0.75 mg/mL, CH_2_Cl_2_); **IR (ν_max_/cm^−1^)** 2936.05 (CH str.), 1708.51 (4 × C=O); **^1^H NMR (300 MHz, CDCl_3_)** δ_H_ 7.77 (d, *J* = 7.4 Hz, 1H, CH_aro_), 7.62–7.54 (m, 2H, 2 × CH_aro_), 7.53–7.45 (m, 1H, CH_aro_), 7.40–7.27 (m, 9H, 9 × CH_aro_), 7.25–7.18 (m, 1H, CH_aro_), 6.53 (d, *J* = 18.8 Hz, 1H, CH_aro_), 5.27 (t, *J* = 3.7 Hz, 1H, CH=C), 5.12–5.01 (m, 2H, OCH_2_Ph), 4.88 (qd, *J* = 17.43, 17.43, 17.42, 2.38 Hz, 2H, TrzCH_2_CO_2_), 4.52–4.43 (m, 1H, CHCO_2_), 4.28–4.10 (m, 2H, CH_3_C**H_2_**O), 4.03–3.93 (m, 1H, CCH_2_^b^Trz), 3.87–3.77 (m, 1H, CCH_2_^a^Trz), 3.67 (t, *J* = 7.9 Hz, 2H, NC**H_2_**CH_2_Ph), 3.18–2.98 (m, 2H, NCH_2_C**H_2_**Ph), 2.90 (dd, *J* = 13.7, 4.5 Hz, 1H, C=CC**H**CH_2_), 2.05–1.73 (m, 4H, 2 × CH_2_), 1.71–1.52 (m, 8H, 4 × CH_2_), 1.49–1.24 (m, 8H, 4 × CH_2_), 1.20 (t, *J* = 7.1 Hz, 3H, C**H_3_**CH_2_O), 1.16 (d, *J* = 8.6 Hz, 1H, CH), 1.10 (s, 3H, CH_3_), 1.00 (d, *J* = 3.9 Hz, 1H, CH), 0.93–0.84 (m, 9H, 3 × CH_3_), 0.76 (s, 3H, CH_3_), 0.68 (s, 3H, CH_3_), 0.59 (s, 3H, CH_3_); **^13^C NMR (75 MHz,CDCl_3_)**δ_C_ 177.55 (C=O)_,_ 170.31 (C=O), 169.22 (C=O), 165.83 (NC=O), 143.86 (C_q_), 143.28 (C_q_), 142.99 (C_q_), 140.92 (C_q_), 139.24 (C_q_), 136.54 (2 × C_q_), 132.29 (CH_aro_), 129.61 (CH_aro_), 129.05 (2 × CH_aro_), 128.68 (2 × CH_aro_), 128.54 (2 × CH_aro_), 128.11 (2 × CH_aro_), 128.04 (CH_aro_), 126.54 (CH_aro_), 123.81 (CH_aro_), 122.84 (CH_aro/Trz_), 122.41 (CH=C), 122.24 (CH_aro_), 83.79 (CHCO_2_), 71.45 (C_q_), 66.05 (OCH_2_Ph), 62.77 (CH_3_**C**H_2_O), 55.27 (CH), 51.06 (TrzCH_2_CO_2_), 47.60 (CH), 46.85 (C_q_), 45.96 (CH_2_), 44.25 (N**C**H_2_CH_2_Ph), 41.79 (C_q_), 41.48 (C=C**C**HCH_2_), 39.37 (C_q_), 38.06 (CH_2_), 37.83 (C_q_), 36.94 (C_q_), 34.36 (NCH_2_C**H_2_**Ph), 33.97 (CH_2_), 33.22 (CH_3_), 32.68 (CH_2_), 32.47 (CH_2_), 30.82 (C_q_), 30.69 (C**C**H_2_Trz), 28.17 (CH_3_), 27.71 (CH_2_), 25.96 (CH_3_), 23.77 (CH_3_), 23.48 (2 × CH_2_), 23.15 (CH_2_), 18.25 (CH_2_), 16.96 (CH_3_), 16.65 (CH_3_), 15.39 (CH_3_), 14.09 (**C**H_3_CH_2_O); HRMS (+ESI) calculated for C_61_H_77_N_4_O_7_ [M + H]^+^: 977.5748, found: 977.5807.

*Methyl2-allyl-1-((1-(2-(((3S,4aR,6aR,6bS,8aS,12aS,14aR,14bR)-8a-((benzyloxy)carbonyl)-4,4,6a,6b,11,11,14b-heptamethyl-1,2,3,4,4a,5,6,6a,6b,7,8,8a,9,10,11,12,12a,14,14a,14b-icosahydropicen-3-yl)oxy)-2-oxoethyl)-1H-1,2,3-triazol-4-yl)methyl)-3-oxoisoindoline-1-carboxylate* (**18n**), This derivative was isolated as a white solid in 70% yield; Rf (cyclohexane/EtOAc: 60/40) = 0.4; mp = 98–100 °C; [α]_D_^20^ +80 (*c* 0.7 mg/mL, CH_2_Cl_2_); **IR (ν_max_/cm^−1^)** 2935.43 (CH str.), 1710.09 (4 × C=O); **^1^H NMR (300 MHz, CDCl_3_)** δ_H_ 7.75 (d, *J* = 7.4, 1.3 Hz, 1H_,_ CH_aro_), 7.61–7.43 (m, 3H, 3 × CH_aro_), 7.37–7.29 (m, 5H, 5 × CH_aro_), 6.55 (d, *J* = 21.3 Hz, 1H, CH_aro_), 5.98–5.81 (m, 1H, C**H**=CH_2_), 5.36–5.16 (m, 3H, CH=C, CH=C**H_2_**), 5.14–4.99 (m, 2H, OCH_2_Ph), 4.98–4.77 (m, 2H, TrzCH_2_CO_2_), 4.53–4.35 (m, 2H, CHCO_2_, NC**H_2_**^b^CH=CH_2_), 4.11–4.00 (m, 1H, NC**H_2_**^a^CH=CH_2_), 3.88 (qd, *J* = 15.72, 15.71, 15.71, 4.49 Hz, 2H, CCH_2_Trz), 3.65 (s, 3H, CH_3_O), 2.89 (dd, *J* = 13.5, 4.3 Hz, 1H, C=CC**H**CH_2_), 2.03–1.73 (m, 4H, 2 × CH_2_), 1.72–1.62 (m, 4H, 2 × CH_2_), 1.61–1.57 (m, 2H, CH_2_), 1.51–1.43 (m, 2H, CH_2_),1.41–1.18 (m, 8H, 4 × CH_2_), 1.16 (d, *J* = 3.8 Hz, 1H, CH), 1.10 (s, 3H, CH_3_), 1.04–0.99 (m, 1H, CH), 0.94–0.82 (m, 9H, 3 × CH_3_), 0.77 (s, 3H, CH_3_), 0.69 (s, 3H, CH_3_), 0.58 (s, 3H, CH_3_); **^13^C NMR (75 MHz,CDCl_3_)** δ_C_ 177.56 (C=O), 170.80 (C=O), 168.70 (C=O), 165.87 (NC=O), 143.86 (C_q_), 142.38 (C_q_), 140.86 (C_q_), 136.53 (2 × C_q_), 133.11 (**C**H=CH_2_), 132.35 (CH_aro_), 129.61 (CH_aro_), 128.53 (2 × CH_aro_), 128.10 (2 × CH_aro_), 128.04 (CH_aro_), 124.03 (CH_aro_), 122.91 (CH_aro/Trz_), 122.40 (**C**H=C), 121.93 (CH_aro_) 118.54 (CH=**C**H_2_), 83.80 (CHCO_2_), 70.35 (C_q_), 66.04 (OCH_2_Ph), 55.28 (CH), 53.10 (CH_3_O), 51.04 (TrzCH_2_CO_2_), 47.60 (CH), 46.84 (C_q_), 45.95 (CH_2_), 44.15 (N**C**H_2_CH=CH_2_), 41.78 (C_q_), 41.47 (C=C**C**HCH_2_), 39.37 (C_q_), 38.07 (CH_2_), 37.86 (C_q_), 36.94 (C_q_), 33.96 (CH_2_), 33.22 (CH_3_), 32.68 (CH_2_), 32.46 (CH_2_), 30.82 (C_q_), 30.02 (C**C**H_2_Trz), 28.16 (CH_3_), 27.70 (CH_2_), 25.95 (CH_3_), 23.76 (CH_3_), 23.47 (2 × CH_2_), 23.14 (CH_2_), 18.25 (CH_2_), 16.95 (CH_3_), 16.64 (CH_3_), 15.42 (CH_3_); HRMS (+ESI) calculated for C_55_H_71_N_4_O_7_ [M + H]^+^: 899.5278, found: 899.5327.

*(4aS,6aS,6bR,8aR,10S,12aR,12bR,14bS)-benzyl 10-(2-(4-((2-benzyl-1-((2-methoxy-2-oxoethyl)carbamoyl)-3-oxoisoindolin-1-yl)methyl)-1H-1,2,3-triazol-1-yl)acetoxy)-2,2,6a,6b,9,9,12a-heptamethyl1,2,3,4,4a,5,6,6a,6b,7,8,8a,9,10,11,12,12a,12b,13,14b-icosahydropicene-4a-carboxylate* (**18o**), This derivative was isolated as a white solid in 94% yield; Rf (cyclohexane/EtOAc: 40/60) = 0.5; mp = 120–122 °C; [α]_D_^20^ +41 (*c* 1 mg/mL, CH_2_Cl_2_); **IR (ν_max_/cm^−1^)** 3300 (NH), 2931.02 (CH str.), 1726.01 (3 × C=O), 1677.77 (C=O); **^1^H NMR (300 MHz, CDCl_3_)** δ_H_ 7.72 (d, *J* = 7.4 Hz, 1H_,_ CH_aro_), 7.63–7.39 (m, 6H_,_ 6 × CH_aro_), 7.36–7.27 (m, 6H_,_ 6 × CH_aro_), 7.25–7.19 (m, 1H_,_ CH_aro_), 6.32 (d, *J* = 11.7 Hz, 1H, CH_aro_), 5.86 (br. s, 1H, NH), 5.27 (t, *J* = 3.8 Hz, 1H, CH=C), 5.12–4.76 (m, 5H, NCH_2_^b^Ph,TrzCH_2_CO_2_, OCH_2_Ph), 4.65 (dd, *J* = 15.2, 6.7 Hz, 1H, NCH_2_^a^Ph), 4.55–4.44 (m, 1H, CHCO_2_), 3.96 (d, *J* = 15.6 Hz, 1H, CCH_2_^b^Trz), 3.85–3.56 (m, 5H, CCH_2_^a^Trz, NHCH_2_^b^CO_2_, CH_3_O), 3.28–3.14 (m, 1H, NHCH_2_^a^CO_2_), 2.89 (dd, *J* = 13.8, 4.4 Hz, 1H, C=CC**H**CH_2_), 2.04–1.79 (m, 4H, 2 × CH_2_), 1.69–1.63 (m, 2H, CH_2_), 1.52–1.46 (m, 2H, CH_2_), 1.44–1.31 (m, 4H, 2 × CH_2_), 1.29–1.19 (m, 8H, 4 × CH_2_), 1.16 (d, *J* = 4.0 Hz, 1H, CH), 1.10 (s, 3H, CH_3_), 1.04–1.00 (m, 1H, CH), 0.92–0.87 (m, 9H, 3 × CH_3_), 0.78 (s, 3H, CH_3_), 0.68 (s, 3H, CH_3_), 0.59 (s, 3H, CH_3_); **^13^C NMR (75 MHz,CDCl_3_)** δ_C_ 177.55 (C=O), 169.83 (NHC=O)_,_ 169.18 (2 × C=O), 165.83 (NC=O)_,_ 144.31 (C_q_), 143.85 (C_q_), 141.05 (C_q_), 137.54 (C_q_), 136.52 (C_q_), 132.81 (CH_aro_), 130.95 (C_q_), 129.66 (CH_aro_), 129.42 (2 × CH_aro_), 128.83 (2 × CH_aro_), 128.52 (2 × CH_aro_), 128.09 (2 × CH_aro_), 128.02 (CH_aro_), 127.83 (CH_aro_), 124.19 (CH_aro_), 123.54 (CH_aro_), 122.63 (CH_aro/Trz_), 122.39 (CH=C), 83.73 (CHCO_2_), 71.66 (C_q_), 66.04 (OCH_2_Ph), 55.27 (CH), 52.34 (CH_3_O), 50.98 (TrzCH_2_CO_2_), 47.60 (CH), 46.83 (C_q_), 45.94 (CH_2_), 44.84 (NCH_2_Ph), 41.78 (C_q_), 41.46 (C=C**C**HCH_2_), 41.29 (NH**C**H_2_CO_2_), 39.36 (C_q_), 38.07 (CH_2_), 37.86 (C_q_), 36.94 (C_q_), 33.95 (CH_2_), 33.20 (CH_3_), 32.67 (CH_2_), 32.46 (CH_2_), 30.81 (C_q_), 29.81 (CCH_2_^b^Trz), 28.16 (CH_3_), 27.69 (CH_2_), 25.95 (CH_3_), 23.75 (CH_3_), 23.47 (2 × CH_2_), 23.13 (CH_2_), 18.24 (CH_2_), 16.95 (CH_3_), 16.64 (CH_3_), 15.41 (CH_3_); HRMS (+ESI) calculated for C_61_H_75_N_5_O_8_Na[M+Na]^+^: 1028.5508, found: 1028.5584.

*(4aS,6aS,6bR,8aR,10S,12aR,12bR,14bS)-benzyl 10-(2-(4-((2-((1,5-dimethyl-1H-pyrrol-2-yl)methyl)-1-(hydroxymethyl)-3-oxoisoindolin-1-yl)methyl)-1H-1,2,3-triazol-1-yl)acetoxy)-2,2,6a,6b,9,9,12a-heptamethyl-1,2,3,4,4a,5,6,6a,6b,7,8,8a,9,10,11,12,12a,12b,13,14b-icosahydropicene-4a-carboxylate* (**18p**), This derivative was isolated as a yellow solid in 70% yield; Rf (cyclohexane/EtOAc: 40/60) = 0.5; mp = 85–87 °C; [α]_D_^20^ +64 (*c* 0.85 mg/mL, CH_2_Cl_2_); **IR (ν_max_/cm^−1^)** 3400 (OH), 2923.75 (CH str.), 1726.83 (C=O), 1678.10 (2 × C=O); **^1^H NMR (300 MHz, CDCl_3_) **δ_H_ 7.78 (d, *J* = 7.5 Hz, 1H, CH_aro_), 7.63–7.40 (m, 3H, 3 × CH_aro_), 7.39–7.28 (m, 6H, 6 × CH_aro_), 6.44 (d, *J* = 16.9 Hz, 1H_,_ CH_aro_), 6.16 (t, *J* = 2.6 Hz, 1H, CH_aro_), 5.84 (d, *J* = 3.3 Hz, 1H, CH_aro_), 5.27 (t, *J* = 3.7 Hz, 1H, CH=C), 5.19–4.79 (m, 5H, NCH_2_^b^Hyd, TrzCH_2_CO_2_, OCH_2_Ph), 4.64–4.45 (m, 2H, NCH_2_^a^Hyd, CHCO_2_), 3.82–3.70 (m, 2H, CH_2_OH), 3.53–3.36 (m, 5H, CH_3_N, CCH_2_Trz), 2.89 (dd, *J* = 14.0, 4.6 Hz, 1H, C=CC**H**CH_2_), 2.18 (s, 3H, NCCH_3_), 1.99–1.81 (m, 4H, 2 × CH_2_), 1.74–1.65 (m, 2H, CH_2_), 1.59–1.56 (m, 2H, CH_2_), 1.37–1.19 (m, 8H, 4 × CH_2_), 1.16 (d, *J* = 2.8 Hz, 1H, CH), 1.11 (s, 3H,CH_3_), 1.54–1.39 (m, 4H, 2 × CH_2_), 1.05–1.00 (m, 1H, CH), 0.93–0.86 (m, 9H, 3 × CH_3_), 0.79 (s, 3H, CH_3_), 0.69 (s, 3H, CH_3_), 0.59 (s, 3H, CH_3_); **^13^C NMR (75 MHz,CDCl_3_)** δ_C_ 177.55 (C=O), 168.51 (C=O), 165.93 (NC=O), 146.14 (C_q_), 143.86 (C_q_), 141.70 (C_q_), 136.53 (C_q_), 132.29 (C_q_), 132.10 (CH_aro_), 130.78 (C_q_), 128.90 (CH_aro_), 128.52 (2 × CH_aro_), 128.09 (2 × CH_aro_), 128.03 (CH_aro_), 127.19 (C_q_), 123.92 (CH_aro_), 123.09 (CH_aro/Trz_), 122.39 (CH=C), 121.92 (CH_aro_), 108.56 (CH_aro_), 105.93 (CH_aro_), 83.83 (CHCO_2_), 70.12 (C_q_), 66.72 (CH_2_OH), 66.04 (OCH_2_Ph), 55.29 (CH), 51.05 (TrzCH_2_CO_2_), 47.60 (CH), 46.84 (C_q_), 45.96 (CH_2_), 41.79 (C_q_), 41.47 (C=C**C**HCH_2_), 39.38 (C_q_), 38.08 (CH_2_), 37.87 (C_q_), 36.95 (C_q_), 35.98 (NCH_2_Hyd), 33.96 (CH_2_), 33.21 (CH_3_), 32.68 (CH_2_), 32.46 (CH_2_), 30.88 (CH_3_N), 30.81 (C_q_), 29.82 (CCH_2_Trz), 28.19 (CH_3_), 27.71 (CH_2_), 25.96 (CH_3_), 23.75 (CH_3_), 23.49 (2 × CH_2_), 23.14 (CH_2_), 18.26 (CH_2_), 16.96 (CH_3_), 16.67 (CH_3_), 15.44 (CH_3_), 12.66 (NC**C**H_3_); HRMS (+ESI) calculated for C_58_H_75_N_5_O_6_Na [M+Na]^+^: 960.5610, found: 960.5708.

*1-((1-(2-(((3S,4aR,6aR,6bS,8aS,12aS,14aR,14bR)-8a-((benzyloxy)carbonyl)-4,4,6a,6b,11,11,14b-heptamethyl-1,2,3,4,4a,5,6,6a,6b,7,8,8a,9,10,11,12,12a,14,14a,14b-icosahydropicen-3-yl)oxy)-2-oxoethyl)-1H-1,2,3-triazol-4-yl)methyl)-2-(furan-2-ylmethyl)-3-oxoisoindoline-1-carboxylic acid* (**18q**), This derivative was isolated as a yellow solid in 60% yield; Rf (cyclohexane/EtOAc: 60/40) = 0.4; mp = 81–83 °C; [α]_D_^20^ +76 (*c* 0.5 mg/mL, CH_2_Cl_2_); **IR (ν_max_/cm^−1^)** 2924.69 (CH str.), 1724.90 (3 × C=O), 1711 (C=O); **^1^H NMR (300 MHz, CDCl_3_)** δ_H_ 7.79 (d, *J* = 7.4 Hz, 1H_,_ CH_aro_), 7.55–7.27 (m, 10H, 10 × CH_aro_), 6.34 (dd, *J* = 6.9, 2.8 Hz, 2H, 2 × CH_aro_), 5.27 (t, *J* = 4.2 Hz, 1H, CH=C), 5.23–4.78 (m, 6H, NCH_2_Fur, TrzCH_2_CO_2_, OCH_2_Ph), 4.58–4.35 (m, 2H, CHCO_2_, CCH_2_^b^Trz), 3.51 (s, 1H, CCH_2_^a^Trz), 2.89 (dd, *J* = 14.1, 4.5 Hz, 1H, C=CC**H**CH_2_), 2.05–1.80 (m, 4H, 2 × CH_2_), 1.73–1.61 (m, 4H, 2 × CH_2_), 1.57–1.44 (m, 4H, 2 × CH_2_), 1.43–1.19 (m, 8H, 4 × CH_2_), 1.16 (d, *J* = 3.7 Hz, 1H, CH), 1.10 (s, 3H, CH_3_), 1.05–1.00 (m, 1H, CH), 0.93–0.86 (m, 9H, 3 × CH_3_), 0.80 (s, 3H, CH_3_), 0.72 (s, 3H, CH_3_), 0.59 (s, 3H, CH_3_); **^13^C NMR (75 MHz,CDCl_3_)** δ_C_ 177.53 (C=O), 171.42 (C=O), 168.26 (C=O), 165.83 (NC=O), 150.45 (C_q_), 144.47 (C_q_), 143.85 (C_q_), 142.66 (CH_aro_), 136.51 (2 × C_q_), 132.25 (C_q_), 131.88 (CH_aro_), 128.56 (2 × CH_aro_), 128.51 (2 × CH_aro_), 128.08 (2 × CH_aro_), 128.02 (CH_aro_), 123.91 (CH_aro_), 122.68 (CH_aro/Trz_), 122.37 (CH=C), 110.65 (CH_aro_), 108.94 (CH_aro_), 83.87 (CHCO_2_), 68.76 (C_q_), 66.02 (OCH_2_Ph), 55.28 (CH), 51.71 (TrzCH_2_CO_2_),47.59 (CH), 46.82 (C_q_), 45.94 (CH_2_), 41.77, (C_q_), 41.46 (C=C**C**HCH_2_), 39.36 (C_q_), 38.07 (CH_2_), 37.86 (C_q_), 37.06 (NCH_2_Fur), 36.94 (C_q_), 33.94 (CH_2_), 33.20 (CH_3_), 32.66 (CH_2_), 32.44 (CH_2_), 30.80 (C_q_), 29.80 (C**C**H_2_Trz), 28.18 (CH_3_), 27.69 (CH_2_), 25.94 (CH_3_), 23.74 (CH_3_), 23.48 (2 × CH_2_), 23.12 (CH_2_), 18.24 (CH_2_), 16.94 (CH_3_), 16.65 (CH_3_), 15.42 (CH_3_); HRMS (+ESI) calculated for C_56_H_69_N_4_O_8_[M + H]^+^: 925.5071, found: 925.5109.

*Ethyl2-((1-(2-(((3S,4aR,6aR,6bS,8aS,12aS,14aR,14bR)-8a-((benzyloxy)carbonyl)-4,4,6a,6b,11,11,14b-heptamethyl-1,2,3,4,4a,5,6,6a,6b,7,8,8a,9,10,11,12,12a,14,14a,14b-icosahydropicen-3-yl)oxy)-2-oxoethyl)-1H-1,2,3-triazol-4-yl)methyl)-1-(2-ethoxy-2-oxoethyl)-3-oxoisoindoline-1-carboxylate* (**18r**), This derivative was isolated as a white solid in 98% yield; Rf (cyclohexane/EtOAc: 60/40) = 0.5; mp = 97–99 °C; [α]_D_^20^ +55 (*c* 1 mg/mL, CH_2_Cl_2_); **IR (ν_max_/cm^−1^)** 2936.99 (CH str.), 1710.10 (5 × C=O); **^1^H NMR (300 MHz, CDCl_3_)**δ_H_ 7.88–7.77 (m, 2H, 2 × CH_aro_), 7.61–7.45 (m, 3H, 3 × CH_aro_), 7.39–7.28 (m, 5H, 5 × CH_aro_), 5.27 (t, *J* = 3.7 Hz, 1H, CH=C), 5.15–4.85 (m, 6H, NCH_2_Trz, TrzCH_2_CO_2_, OCH_2_Ph), 4.53 (dd, *J* = 9.5, 6.5 Hz, 1H, CHCO_2_), 4.26–4.05 (m, 2H, CH_3_C**H_2_**CO_2_C), 4.03–3.92 (m, 2H, CH_3_C**H_2_**CO_2_CH_2_), 3.52 (d, *J* = 17.1 Hz, 1H, CCH_2_^b^CO_2_), 3.13 (dd, *J* = 17.0, 2.6 Hz, 1H, CCH_2_^a^CO_2_), 2.89 (dd, *J* = 14.2, 4.4 Hz, 1H, C=CC**H**CH_2_), 2.01–1.80 (m, 4H, 2 × CH_2_), 1.78–1.63 (m, 4H, 2 × CH_2_), 1.62–1.54 (m, 4H, 2 × CH_2_), 1.54–1.27 (m, 8H, 4 × CH_2_), 1.25 (d, *J* = 4.6 Hz, 1H, CH), 1.19 (td, *J* = 7.1, 2.5 Hz, 3H, C**H_3_**CH_2_CO_2_CH_2_), 1.11–1.03 (m, 6H, C**H_3_**CH_2_CO_2_C, CH_3_), 1.02–0.98 (m, 1H, CH), 0.92–0.83 (m, 9H, 3 × CH_3_), 0.80 (s, 3H, CH_3_), 0.69 (s, 3H, CH_3_), 0.58 (s, 3H, CH_3_); **^13^C NMR (75 MHz,CDCl_3_)** δ_C_ 177.55 (C=O), 169.08 (C=O), 169.40 (C=O), 168.92 (C=O), 165.91 (NC=O), 143.85 (C_q_), 143.57 (C_q_), 136.54 (2 × C_q_), 132.40 (2 × CH_aro_), 131.20 (C_q_), 129.52 (CH_aro_), 128.53 (2 × CH_aro_), 128.09 (2 × CH_aro_), 128.03 (CH_aro_), 123.74 (CH_aro_), 122.40 (CH_aro/Trz_+ **C**H=C), 83.83 (CHCO_2_), 69.23 (C_q_), 66.03 (OCH_2_Ph), 62.77 (CH_3_**C**H_2_CO_2_C), 61.12 (CH_3_**C**H_2_CO_2_CH_2_), 55.28 (CH), 51.17 (TrzCH_2_CO_2_), 47.59 (CH), 46.84 (C_q_), 45.95 (CH_2_), 41.78 (C_q_), 41.47 (C=C**C**HCH_2_), 40.38 (CCH_2_^b^CO_2_), 39.37 (C_q_), 38.07 (CH_2_), 37.83 (C_q_), 36.94 (NCH_2_Trz),36.92 (C_q_), 33.96 (CH_2_), 33.21 (CH_3_), 32.67 (CH), 32.46 (CH_2_), 30.81 (C_q_), 28.18 (CH_3_), 27.70 (CH_2_), 25.95 (CH_3_), 23.76 (CH_3_), 23.48 (2 × CH_2_), 23.14 (CH_2_), 18.23 (CH_2_), 16.96 (CH_3_), 16.63 (CH_3_), 15.40 (CH_3_), 14.00 (**C**H_3_CH_2_CO_2_C, **C**H_3_CH_2_CO_2_CH_2_); HRMS (+ESI) calculated for C_57_H_75_N_4_O_9_ [M + H]^+^: 959.5489, found: 959.5543.

*Ethyl1-allyl-2-((1-(2-(((3S,4aR,6aR,6bS,8aS,12aS,14aR,14bR)-8a-((benzyloxy)carbonyl)-4,4,6a,6b,11,11,14b-heptamethyl-1,2,3,4,4a,5,6,6a,6b,7,8,8a,9,10,11,12,12a,14,14a,14b-icosahydropicen-3-yl)oxy)-2-oxoethyl)-1H-1,2,3-triazol-4-yl)methyl)-3-oxoisoindoline-1-carboxylate* (**18s**), This derivative was isolated as a white solid in 90% yield; Rf (cyclohexane/EtOAc: 60/40) = 0.5; mp = 100–102 °C; [α]_D_^20^ +53 (*c* 1.5 mg/mL, CH_2_Cl_2_); **IR (ν_max_/cm^−1^)** 2943.31 (CH str.), 1731.34 (2 × C=O), 1701.17 (2 × C=O); **^1^H NMR (300 MHz, CDCl_3_**) δ_H_ 7.93 (s, 1H, CH_aro/Trz_), 7.80 (d, *J* = 7.4 Hz, 1H, CH_aro_), 7.62–7.44 (m, 3H, 3 × CH_aro_), 7.39–7.28 (m, 5H, 5 × CH_aro_), 5.26 (t, *J* = 3.7 Hz, 1H, CH=C), 5.17–4.67 (m, 9H, NCH_2_Trz, TrzCH_2_CO_2_, OCH_2_Ph, C**H**=CH_2_, CH=C**H_2_**), 4.55 (t, *J* = 8.0 Hz, 1H, CHCO_2_), 4.18–4.01 (m, 2H, CH_3_C**H_2_**O), 3.26–3.11 (m, 2H, C**H_2_**CH=CH_2_), 2.89 (dd, *J* = 13.6, 4.4 Hz, 1H, C=CC**H**CH_2_), 2.05–1.78 (m, 6H, 3 × CH_2_), 1.75–1.62 (m, 4H, 2 × CH_2_),1.57–1.50 (m, 2H, CH_2_), 1.46–1.23 (m, 8H, 4 × CH_2_), 1.20 (d, *J* = 3.1 Hz, 1H, CH), 1.14 (dt, *J* = 7.1, 3.5 Hz, 3H, C**H_3_**CH_2_O), 1.10 (s, 3H, CH_3_), 1.04–0.99 (m, 1H, CH), 0.93–0.85 (m, 9H, 3 × CH_3_), 0.82 (s, 3H, CH_3_), 0.73 (s, 3H, CH_3_), 0.58 (s, 3H, CH_3_); **^13^C NMR (75 MHz,CDCl_3_)** δ_C_ 177.58 (C=O), 169.97 (C=O), 169.28 (C=O), 165.87 (NC=O), 144.46 (C_q_), 143.85 (C_q_), 143.60 (C_q_), 136.55 (C_q_), 132.35 (**C**H=CH_2_), 131.49 (C_q_), 129.82 (CH_aro_), 129.27 (CH_aro_), 128.54 (2 × CH_aro_), 128.10 (2 × CH_aro_), 128.04 (CH_aro_), 125.88 (CH_aro_), 123.69 (CH_aro_), 122.42 (CH_aro/Trz_, CH=C), 120.29 (CH=**C**H_2_), 83.89 (CHCO_2_), 72.31 (C_q_), 66.05 (OCH_2_Ph), 62.57 (CH_2_), 55.30 (CH), 51.23 (TrzCH_2_CO_2_), 47.60 (CH), 46.85 (C_q_), 45.95 (CH_2_), 41.79 (C_q_), 41.48 (C=C**C**HCH_2_), 39.38 (C_q_), 38.09 (CH_2_), 37.86 (C_q_), 37.77 (**C**H_2_CH=CH_2_), 37.02 (NCH_2_Trz), 36.96 (C_q_), 33.97 (CH_2_), 33.22 (CH_3_), 32.69 (CH_2_), 32.48 (CH_2_), 30.82 (C_q_), 28.22 (CH_3_), 27.71 (CH_2_), 25.96 (CH_3_), 23.77 (CH_3_), 23.51 (2 × CH_2_), 23.15 (CH_2_), 18.25 (CH_2_), 16.97 (CH_3_), 16.69 (CH_3_), 15.43 (CH_3_), 14.03 (**C**H_3_CH_2_O); HRMS (+ESI) calculated for C_56_H_73_N_4_O_7_ [M + H]^+^: 913.5435, found: 913.5466.

*Ethyl2-((1-(2-(((3S,4aR,6aR,6bS,8aS,12aS,14aR,14bR)-8a-((benzyloxy)carbonyl)-4,4,6a,6b,11,11,14b-heptamethyl-1,2,3,4,4a,5,6,6a,6b,7,8,8a,9,10,11,12,12a,14,14a,14b-icosahydropicen-3-yl)oxy)-2-oxoethyl)-1H-1,2,3-triazol-4-yl)methyl)-1-(3-methoxybenzyl)-3-oxoisoindoline-1-carboxylate* (**18t**), This derivative was isolated as a white solid in 79% yield; Rf (cyclohexane/EtOAc: 60/40) = 0.5; mp = 100–102 °C; [α]_D_^20^ +70 (*c* 1.5 mg/mL, CH_2_Cl_2_); **IR (ν_max_/cm^−1^)** 2936.46 (CH str.), 1702.26 (4 × C=O); **^1^H NMR (300 MHz, CDCl_3_)** δ_H_ 7.75, (s, 1H, CH_aro/Trz_), 7.69 (d, *J* = 7.5 Hz, 1H,CH_aro_), 7.63–7.53 (m, 2H, 2 × CH_aro_), 7.50–7.42 (m, 1H, CH_aro_), 7.40–7.28 (m, 5H, 5 × CH_aro_), 6.86 (td, *J* = 7.9, 2.5 Hz, 1H, CH_aro_), 6.56 (d, *J* = 7.8 Hz, 1H, CH_aro_), 6.16 (t, *J* = 6.3 Hz, 1H, CH_aro_), 5.99 (s, 1H, CH_aro_), 5.27 (t, *J* = 3.9 Hz, 1H, CH=C), 5.10–4.85 (m, 4H, TrzCH_2_CO_2_, OCH_2_Ph), 4.86 (s, 2H, NCH_2_Trz), 4.54 (t, *J* = 7.9 Hz, 1H, CHCO_2_), 4.12–4.01 (m, 2H, CH_3_C**H_2_**O), 3.83 (d, *J* = 14.7 Hz, 1H, CC**H_2_**^b^3-OMePh), 3.57 (d, *J* = 14.7 Hz, 1H, CC**H_2_**^a^3-OMePh), 3.49 (s, 3H, CH_3_O), 2.89 (dd, *J* = 13.8, 4.4 Hz, 1H, C=CC**H**CH_2_), 2.18–1.72 (m, 4H, 2 × CH_2_), 1.70–1.61 (m, 4H, 2 × CH_2_), 1.58–1.45 (m, 4H, 2 × CH_2_), 1.43–1.18 (m, 8H, 4 × CH_2_), 1.16 (d, *J* = 3.7 Hz, 1H, CH), 1.10 (s, 3H, CH_3_), 1.08–1.02 (m, 3H, C**H_3_**CH_2_O), 1.01–0.97 (m, 1H, CH), 0.93–0.79 (m, 12H, 4 × CH_3_), 0.72 (s, 3H, CH_3_), 0.58 (s, 3H, CH_3_); **^13^C NMR (75 MHz,CDCl_3_)** δ_C_ 177.52 (C=O), 169.96 (C=O), 169.28 (C=O), 165.83 (NC=O), 158.96 (C_q_), 143.95 (C_q_), 143.81 (C_q_), 136.50 (C_q_), 135.19 (C_q_), 132.01 (CH_aro_), 131.55 (2 × C_q_), 129.28 (CH_aro_), 128.86 (CH_aro_), 128.50 (2 × CH_aro_), 128.06 (2 × CH_aro_), 127.99 (CH_aro_), 125.74 (CH_aro_), 123.71 (CH_aro_), 122.89 (CH_aro_), 122.38 (CH=C), 122.17 (CH_aro/Trz_), 114.87 (CH_aro_), 113.00 (CH_aro_), 83.81 (CHCO_2_), 72.80 (C_q_), 66.00 (OCH_2_Ph), 62.64 (CH_3_**C**H_2_O), 55.25 (CH), 54.97 (CH_3_O), 51.14 (TrzCH_2_CO_2_), 47.55 (CH), 46.80 (C_q_), 45.91 (CH_2_), 41.74 (C_q_), 41.43 (C=C**C**HCH_2_), 39.54 (C**C**H_2_3-OMePh), 39.33 (C_q_), 38.04 (CH_2_), 37.81 (C_q_), 37.65 (NCH_2_Trz), 36.90 (C_q_), 33.92 (CH_2_), 33.18 (CH_3_), 32.64 (CH_2_), 32.43 (CH_2_), 30.78 (C_q_), 28.17 (CH_3_), 27.66 (CH_2_), 25.92 (CH_3_), 23.73 (CH_3_), 23.45 (2 × CH_2_), 23.10 (CH_2_), 18.20 (CH_2_), 16.92 (CH_3_), 16.63 (CH_3_), 15.38 (CH_3_), 13.84 (**C**H_3_CH_2_O); HRMS (+ESI) calculated for C_61_H_77_N_4_O_8_ [M + H]^+^: 993.5697, found: 993.5744.

*Ethyl1,2-bis((1-(2-(((3S,4aR,6aR,6bS,8aS,12aR,14aS,14bS)-8a-((benzyloxy)carbonyl)-4,4,6a,6b,11,11,14b-heptamethyl-1,2,3,4,4a,5,6,6a,6b,7,8,8a,9,10,11,12,12a,14,14a,14b-icosahydropicen-3-yl)oxy)-2-oxoethyl)-1H-1,2,3-triazol-4-yl)methyl)-3-oxo-2,3-dihydro-1H-isoindole-1-carboxylate* (**18u**), This derivative was isolated as a white solid in 80% yield; Rf (cyclohexane/EtOAc: 50/50) = 0.4; mp = 143–145 °C; [α]_D_^20^ +63 (*c* 0.9 mg/mL, CH_2_Cl_2_); **IR (ν_max_/cm^−1^)** 2938.94 (CH str.), 1728.35 (6 × C=O); **^1^H NMR (300 MHz, CDCl_3_)**δ_H_ 7.92 (d, *J* = 6.6 Hz, 1H,CH_aro_), 7.73–7.61 (m, 2H, 2 × CH_aro_), 7.46–7.28 (m, 12H, 12 × CH_aro_), 6.31 (d, *J* = 16.1 Hz, 1H,CH_aro_), 5.26 (t, *J* = 3.6 Hz, 2H, 2 × CH=C), 5.24–4.99 (m, 7H, TrzCH_2_^a^CO_2_, TrzCH_2_CO_2_, 2 × OCH_2_Ph), 4.85 (t, *J* = 3.8 Hz, 2H, 2 × CHCO_2_), 4.67 (dd, *J* = 15.8, 4.5 Hz, 1H, TrzCH_2_^a^CO_2_), 4.60–4.33 (m, 2H, NCH_2_Trz), 4.30–4.05 (m, 3H, CCH_2_^b^Trz,CH_3_C**H_2_**O), 3.89 (dd, *J* = 15.3, 4.3 Hz, 1H, CCH_2_^a^Trz), 2.92 (dd, *J* = 13.6, 4.4 Hz, 2H, 2 × C=CC**H**CH_2_), 2.05–1.74 (m, 8H, 4 × CH_2_), 1.72–1.62 (m, 8H, 4 × CH_2_), 1.60–1.51 (m, 8H, 4 × CH_2_), 1.48–1.24 (m, 16H, 8 × CH_2_),1.21 (t, *J* = 3.6 Hz, 3H, C**H_3_**CH_2_O), 1.17–1.14 (m, 2H, 2 × CH), 1.11 (s, 6H, 2 × CH_3_), 1.04–1.00 (m, 2H, 2 × CH), 0.94–0.80 (m, 24H, 8 × CH_3_), 0.77–0.67 (m, 6H, 2 × CH_3_), 0.61 (s, 6H, 2 × CH_3_); **^13^C NMR (75 MHz,CDCl_3_)** δ_C_ 177.55 (2 × C=O), 169.68 (C=O), 169.61 (C=O), 166.61 (C=O), 166.04 (NC=O), 144.52 (C_q_), 143.85 (2 × C_q_), 142.98 (C_q_), 140.25 (C_q_), 136.53 (C_q_), 132.45 (CH_aro_), 130.98 (C_q_), 129.39 (CH_aro_), 128.53 (4 × CH_aro_), 128.09 (4 × CH_aro_), 128.03 (2 × CH_aro_), 126.01 (CH_aro_), 123.51 (2 × CH_aro/Trz_), 123.26 (CH_aro_), 122.38 (2 × CH=C), 84.31 (CHCO_2_), 83.33 (CHCO_2_), 72.40 (C_q_), 66.04 (2 × OCH_2_Ph), 62.92 (CH_3_**C**H_2_O), 55.24 (2 × CH), 51.03 (2 × TrzCH_2_CO_2_), 47.59 (2 × CH), 46.84 (2 × C_q_), 45.97 (2 × CH_2_), 41.80 (2 × C_q_), 41.48 (2 × C=C**C**HCH_2_), 39.37 (2 × C_q_), 38.08 (2 × CH_2_), 37.74 (C_q_), 37.23 (NCH_2_Trz), 36.95 (2 × C_q_), 33.96 (2 × CH_2_), 33.23 (2 × CH_3_), 32.68 (2 × CH_2_), 32.46 (2 × CH_2_), 30.81 (2 × C_q_), 30.11 (C**C**H_2_Trz), 28.20 (2 × CH_3_), 27.72 (2 × CH_2_), 25.96 (2 × CH_3_), 23.76 (2 × CH_3_), 23.50 (4 × CH_2_), 23.14 (2 × CH_2_), 18.27 (2 × CH_2_), 16.98 (2 × CH_3_), 16.63 (2 × CH_3_), 15.46 (2 × CH_3_), 14.10 (**C**H_3_CH_2_O); HRMS (+ESI) calculated for C_95_H_125_N_7_O_11_Na [M+Na]^+^: 1562.9329, found: 1562.9335.

All ^1^H, ^13^C NMR and HRMS spectra are provided as Appendix A.

### 3.3. Antibacterial Activity

In the present study, the antimicrobial activity of the starting product **OA-1** and its derivatives **18a**–**u** was screened by the agar disc diffusion method according to the protocol described by Dbeibia et al. (2022) [88] against four bacteria, namely *Staphylococcus aureus* ATCC 25923, *Listeria monocytogenes* ATCC 19115, *Salmonella thyphimurium* ATCC 14,080, and *Pseudomonas aeruginosa* ATCC 27853. The inoculums of the microorganisms were adjusted to 0.1 at OD600 and then streaked onto Muller Hinton (MH) agar plates using a sterile cotton mop. Sterile filter discs (diameter 6 mm, Biolife Italy) were placed at the surface of the appropriate agar mediums and 20 µL of the product (10 mg/mL) was dropped onto each disc. Tetracycline and chlorhexidine (10 mg/mL; 20 μL/disc) were used as reference antibiotics. After incubation at 37 °C for 24 h, the antibacterial activities were evaluated by measuring an inhibition zone formed around the disc. Each assay was performed in triplicate. The minimum inhibitory concentration (MIC) was evaluated as recommended by Dbeibia et al. (2022) [88]. Briefly, serial dilutions of the synthesized compounds (3.9–2000 µg/mL) were filled in 96 U bottomed-wells plates (Nunc, Roskilde, Denmark) with MH broth and the target bacteria. The treated plates were left to incubateovernight at 37 °C. The MIC was reported as the lowest concentration of the sample that did not allow the growth of microorganisms and do not show visible turbidity of the broth medium. The minimum bactericidal concentration (MBC) was evaluated by transferring 10 µL from the well showing no bacteria growth after MIC assay, on MH agar. After a 24 h period of incubation at 37 °C, the bacterial growth was examined and the MBC was determined as the lowest concentration of the sample having bactericidal activity.

### 3.4. Molecular Docking Procedure

Molecular docking studies were performed by using Auto Dock 4.2 program package [89]. The optimization of all the geometries of ligands was carried out by ACD (3D viewer) software (http://www.filefacts.com/acd3d-viewer-freeware-info (accessed on 18 January 2023)) and the three dimensional structure of PDB (PDB: 5F7V) [87] was obtained from the RSCB protein data bank (https://www.rcsb.org/ (accessed on 18 January 2023)). Before docking, the water molecules have been erased and the missing hydrogens in additionto Gasteiger charges were then added to the system during the receptor preparation input file. Then, the AutoDock Tools were used for the preparation of all ligands and protein files (PDBQT). The pre-calculation of grid maps was performed by Auto Grid for saving a lot of time during the docking procedure and the docking calculation was carried out by a grid per map with 40 × 40 × 40 A˚ points ofall PDB used in additionto the grid-point spacing of 0.375 A˚, that was centered on the receptor structure in order to determine the active site andthe visualization and analysis of interactions were performed using Discovery Studio 2017R2 (https://www.3dsbiovia.om/products/collaborative-science/biovia-discovery-studio/, accessed on 5 June 2023). Further, all molecular docked models for the cavities 3D were prepared via PyMOL viewer v. 0.99 [90].

### 3.5. Statistical Analysis

Statistical analysis was carried out using Graph Pad Prism 7.0 (Graph Pad Software Inc., CA, USA). The experimental data of the antibacterial activity expressed in the inhibition zone are presented as mean ± standard error of the mean (SEM). Student’s *t*-test was used to assess the difference between two groups. For significant differences among three or more groups, one-way ANOVA with post hoc analysis was performed.

## 4. Conclusions

In summary, oleanolic acid (**OA-1**) isolated from olive pomace (*Olea europaea* L.) was used as a starting material to prepare a series of new (**OA-1**)-phtalimidines coupled 1,2,3-triazole derivatives by application of the Cu(I)-catalyzed Huisgen 1,3-dipolar cycloaddition reaction. The designed cycloadducts were assessed for their antibacterial activity against two Gram-positive (*S. aureus* and *L. monocytogenes*) and two Gram-negative (*S. thyphimurium* and *P. Aeruginosa*) strains. Significant antibacterial activities were observed notably against *L. Monocytogenes*. Interestingly, the derivative **18g** exhibited the highest activity toward this strain (MIC = 9.48 µmol/L) followed by **18d** (MIC = 9.56 µmol/L) and **18h** (MIC = 9.89 µmol/L), compared to tetracycline and chlorhexidine (reference antibiotics). The in silico molecular docking study revealed that the selected lead compounds **18c**, **18d**, **18h,** and **18k** can fit well into the binding cavity of the ABC substrate-binding protein Lmo0181 from *L. monocytogenes*. These results constitute a basis for future works of optimization and expansion of the antibacterial spectrum, especially of derivatives of (**OA-1**)-phthalimidines against Gram-positive bacteria. This also makes it possible to better understand the mechanism of action and the resistance potential of these strains against this new series of semi-synthetic compounds.

## Data Availability

The source data underlying tables and figures are available from the authors upon request.

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
