# Peer review of "Novel Oleanolic Acid-Phtalimidines Tethered 1,2,3 Triazole Hybrids as Promising Antibacterial Agents: Design, Synthesis, In Vitro Experiments and In Silico Docking Studies"

_molecules, 2023, doi:10.3390/molecules28124655_

Round 1

Reviewer 1 Report

the manuscript entitled “Novel Oleanolic acid-Phtalimidines tethered 1,2,3 triazole hybrids as promising antibacterial agents: Design, Synthesis, in vitro experiments and in silico docking studies” (Manuscript ID: 2432047) describes the synthesis of a series of structurally novel oleanolic acid ((3β-hydroxyolean-12-en-28-oic acid, OA-1)-phtalimidines conjugates 18a-v bearing 1,2,3-triazole moieties, which were evaluated for their antibacterial activity against two Gram-positive (Staphylococcus aureus and Listeria monocytogenes) and two Gram-negative (Salmonella thyphimurium and Pseudomonas aeruginosa) bacteria, using Tetracycline as reference drug. Three of these compounds (18g, 18d and 18h) showed significant antibacterial activity against L. Monocytogenes, with MIC values ranging from 9.48 to 9.89 µM. In silico molecular docking studies were performed to give a deeper insight into the interactions established by selected compounds into the active site of the ABC substrate-binding protein Lmo0181 from L. monocytogenes.

Although the results of this study are quite interesting, there are some issues that should be addressed before publication.

·         The authors should clearly explain the choice of ABC substrate-binding protein Lmo0181 as a putative target responsible of the antibacterial activity against Listeria monocytogenes displayed by the tested compounds. Without further explanation it is quite difficult to understand why they performed a computational study focused on this specific target. Moreover, in the abstract section (pag. 1, lines 30-31), the target protein (ABC substrate-binding protein Lmo0181) should be specified.

·         Pag. 2, lines 47-49: in order to highlight the value of nitrogen heterocycles as a valuable tool in medicinal chemistry, the following references should be added (RSC Adv., 2020, 10, 44247-44311-DOI: 10.1039/D0RA09198G; Archives of Pharmacal Research, 2022, 45, 806–DOI: 10.1007/s12272-022-01414-1; Molecules 2020, 25, 1909-DOI:10.3390/molecules25081909). Likewise, in the introduction section (pag. 2, lines 75-78), the authors should emphasize the relevance, in medicinal chemistry, of triazole-based compounds due to their wide range of biological activities. Therefore, the following references should be included (International Journal of Current Research, 2020, 12(08), 12950-12960-DOI: https://doi.org/10.24941/ijcr.39386.08.2020; International Journal of Molecular Sciences, 2023, 24, 7092- DOI: 10.3390/ijms24087092; Current Organic Chemistry, 2022, 26(18), 1691-1702(12)-DOI: https://doi.org/10.2174/1385272827666221213145147; European Journal of Medicinal Chemistry, 2023, 249, 115136-DOI: 10.1016/j.ejmech.2023.115136; Future journal of pharmaceutical sciences (2021), 7(1), 106-DOI: https://doi.org/10.1186/s43094-021-00241-3).

·         Pag. 8, lines 234-236: please clearly point out the structural requirements crucial in conferring high biological activity to the new 1,2,3 triazole hybrid derivatives synthesized.

·         Pag. 8, line 222: please check the numbering of compounds 18g and 18e since they have been switched (compound 18g has the -O-CH2- linker, MIC = 9.48 µM and MBC = 155.65 µM, while compound 18e has the methylene linker directly bonded to the phthalimide nitrogen atom, MIC = 39.01 µM and MBC = 2496.67 µM).

·         Pag. 8, line 230: please correct the MIC value of compound 18c, accordingly to the value reported in the table 2.

·         Pag. 11, lines 289-291: please correct the caption of Figure 3, since there are four images (A, B, C and D) and panel E is missing

·         Pag. 18, lines 619-636: HRMS data of compound 18q are missing.

Other minor points are listed below:

·         Pag. 2, line 50: please replace “are playing” with “play”.

·         Pag. 2, lines 63-64: please replace “with the hope of improving its physical properties for better bioavailability to enhance its bioactivity” with “with the hope of improving its physical properties to obtain better bioavailability and higher bioactivity”.

·         Pag. 2, line 79: please remove the hyphen (“com-pounds”) and also the typos on pag. 7 line 221 (“starin” for “strain”) and on pag. 21 line 778 (“pointfor” for “point for”) and line 781 “thesenewly” for “these newly”)

·         Pag. 7, line 206: please rephrase: “most of the compounds tested proved to be quasi-selective towards L. 206 monocytogenes”.

·         Please carefully check within the manuscript that the names of bacterial strains are reported in italics.

For these reasons, in my opinion, the manuscript is suitable for publishing in Molecules after major revisions.

Overall the quality is satisfying. Minor spelling and grammar check to be done in the revision process.  

Author Response

Pr. Mohamed Othman

Normandie Univ., UNILEHAVRE, FR 3038 CNRS, URCOM  

76600 Le Havre, France.

EA 3221, INC3M CNRS-FR 3038, UFR ST, BP: 1123,

25 rue Philipe Lebon, F-76063 Le Havre Cedex, France

Phone: (+33) 02.32.74.43.98 ; Office fax: (+33) 02.32.74.43.91

E-mail: mohamed.othman@univ-lehavre.fr; othmanlehavre@gmail.com

Le Havre, 2023 May 30

To          Ms. Miranda Yan
Assistant Editor

            Molecules

Manuscript ID: molecules-2432047

Title: Novel Oleanolic acid-Phtalimidines tethered 1,2,3 triazole hybrids as
promising antibacterial agents: Design, Synthesis, in vitro experiments and
in silico docking studies

Authors: Ghofrane Lahmadi, Mabrouk Horchani, Amal Dbeibia, Abdelkarim Mahdhi,
Anis Romdhane, Ata Martin Lawson, Adam Daïch, Abdel Halim Harrath, Hichem
Ben Jannet *, And Mohamed Othman *

Dear colleague,

We would like to express our deep appreciation for your concern regarding the submitted manuscript. We are thankful for the quick and informative reply from the reviewers. We are also grateful for the valuable comments and suggestions from the reviewers aiming to improve the scientific quality of the submitted manuscript.

After reading the reviewers’ comments/suggestions and discussing their comments with other authors of the manuscript, we revised our manuscript and replied to the comments and questions in a point-by-point fashion. Here we enclose the response to the reviewers’ comments.

We will be grateful if the manuscript can be considered for publication in your prestigious journal, Molecules. We deeply appreciate your kind help and concern.

With my very best regards

Sincerely yours,

Pr. Mohamed Othman

Respond to Reviewers’ comments

General remark of the authors to the referees: In order to facilitate the reviewing process, all the modifications we have made in this new version of the paper, including literature citations, appear in yellow and in magenta of the signal assignments in the NMR data.

Answers to referee 1:

Remark 1> The authors should clearly explain the choice of ABC substrate-binding protein Lmo0181 as a putative target responsible of the antibacterial activity against Listeria monocytogenes displayed by the tested compounds. Without further explanation it is quite difficult to understand why they performed a computational study focused on this specific target.

Answer: A paragraph explaining our choice of ABC substrate-binding protein Lmo0181 as a putative target has been added to part: 2.3. Molecular docking study.

Remark 1> Moreover, in the abstract section (pag. 1, lines 30-31), the target protein (ABC substrate-binding protein Lmo0181) should be specified.

Answer: We have included “into the active site of the ABC substrate-binding protein Lmo0181 from L. monocytogenes” to the abstract.

Remark 2> Pag. 2, lines 47-49: in order to highlight the value of nitrogen heterocycles as a valuable tool in medicinal chemistry, the following references should be added (RSC Adv., 2020, 10, 44247-44311-DOI: 10.1039/D0RA09198G; Archives of Pharmacal Research, 2022, 45, 806–DOI: 10.1007/s12272-022-01414-1; Molecules 2020, 25, 1909-DOI:10.3390/molecules25081909). Likewise, in the introduction section (pag. 2, lines 75-78), the authors should emphasize the relevance, in medicinal chemistry, of triazole-based compounds due to their wide range of biological activities. Therefore, the following references should be included (International Journal of Current Research, 2020, 12(08), 12950-12960-DOI: https://doi.org/10.24941/ijcr.39386.08.2020; International Journal of Molecular Sciences, 2023, 24, 7092- DOI: 10.3390/ijms24087092; Current Organic Chemistry, 2022, 26(18), 1691-1702(12)-DOI: https://doi.org/10.2174/1385272827666221213145147; European Journal of Medicinal Chemistry, 2023, 249, 115136-DOI: 10.1016/j.ejmech.2023.115136; Future journal of pharmaceutical sciences (2021), 7(1), 106-DOI: https://doi.org/10.1186/s43094-021-00241-3).

Answer: All references which had been asked by the referee 1 have been included as 15,16,17 and 62,63,64, 65, 66 Refs.

Remark 3> Pag. 8, lines 234-236: please clearly point out the structural requirements crucial in conferring high biological activity to the new 1,2,3 triazole hybrid derivatives synthesized.

Answer: We have enriched this paragraph as suggested.

Remark 4> Pag. 8, line 222: please check the numbering of compounds 18g and 18e since they have been switched (compound 18g has the -O-CH2- linker, MIC = 9.48 µM and MBC = 155.65 µM, while compound 18e has the methylene linker directly bonded to the phthalimide nitrogen atom, MIC = 39.01 µM and MBC = 2496.67 µM).

Answer: Correction of the numbering has been done

Remark 5> Pag. 8, line 230: please correct the MIC value of compound 18c, accordingly to the value reported in the table 2.

Answer: The MIC value of compound 18c was amended.

Remark 6> Pag. 11, lines 289-291: please correct the caption of Figure 3, since there are four images (A, B, C and D) and panel E is missing

AnswerCorrection has been performed in the title of figure 3

Remark 7> Pag. 18, lines 619-636: HRMS data of compound 18q are missing.

Answer: We apologize for this oversight. HRMS data of compound 18q was added.

Remark 8> Pag. 2, line 50: please replace “are playing” with “play”.

Answer: Amended.

 Remark 9> Pag. 2, lines 63-64: please replace “with the hope of improving its physical properties for better bioavailability to enhance its bioactivity” with “with the hope of improving its physical properties to obtain better bioavailability and higher bioactivity”.

Answer: Amended.

Remark 10> Pag. 2, line 79: please remove the hyphen (“com-pounds”) and also the typos on pag. 7 line 221 (“starin” for “strain”) and on pag. 21 line 778 (“pointfor” for “point for”) and line 781 “thesenewly” for “these newly”)

Answer: Corrections have been done

Remark 11> Pag. 7, line 206: please rephrase: “most of the compounds tested proved to be quasi-selective towards L. 206 monocytogenes”.

Answer: Rephrasing has been performed.

Remark 12> Please carefully check within the manuscript that the names of bacterial strains are reported in italics.

Answer: Cheeked and if necessary modified.

Reviewer 2 Report

The article concerns the synthesis of the novel oleanolic acid-phtalimidin estethered 1,2,3 triazole hybrids for antibacterial testing in vitro and in silico docking experiments. The authors have synthesized a series of 21 novel oleanolic acid phtalimidines conjugates marked as 18a-t, and 18v. The substances were adequate characterized all necessary spectral data and screened in vitro for antibacterial activities. The authors have carried out molecular docking study in order to understand the binding mode of the most active substances against the target bacterial protein. The importance of both hydrogen bonding and hydrophobic interactions with the target protein was found and the results were corroborated with the experimental data.

 The article is well written and will be interesting but several corrections are necessary.

      1)      Line 22 and further. The numbering 18a-v is not correct because the substance 18u is absent. The correct numbering is 18a-t, 18v. Fix, please here and in other places.

      2)      Line 26, 192–194, 201 etc. Check and correct the style in Latin names of microorganisms. It should be italic, fix, please in the whole text.

      3)      Line 238. Correct the frames at the tables 1 and 2.

      4)      Line 314. “dd, doublet of doublet” should be replaced with “dd, doublet of doublets”.

      5)      Line 332. Here and further. Delete the spaces around dash in the numbering intervals like in 7.90 – 7.83, i.e. correct on 7.90–7.83.

      6)      Ref. 17. Correct, please the style of the title as in all other references.

      7)      The main flaw of the article is an absence of the signal assignments in the NMR data. Provide, please, the assignment of all the signals for all the  substances and present them in special tables or by any other appropriate way in Supporting materials.

My opinion: minor revision.

The article concerns the synthesis of the novel oleanolic acid-phtalimidin estethered 1,2,3 triazole hybrids for antibacterial testing in vitro and in silico docking experiments. The authors have synthesized a series of 21 novel oleanolic acid phtalimidines conjugates marked as 18a-t, and 18v. The substances were adequate characterized all necessary spectral data and screened in vitro for antibacterial activities. The authors have carried out molecular docking study in order to understand the binding mode of the most active substances against the target bacterial protein. The importance of both hydrogen bonding and hydrophobic interactions with the target protein was found and the results were corroborated with the experimental data.

 The article is well written and will be interesting but several corrections are necessary.

      1)      Line 22 and further. The numbering 18a-v is not correct because the substance 18u is absent. The correct numbering is 18a-t, 18v. Fix, please here and in other places.

      2)      Line 26, 192–194, 201 etc. Check and correct the style in Latin names of microorganisms. It should be italic, fix, please in the whole text.

      3)      Line 238. Correct the frames at the tables 1 and 2.

      4)      Line 314. “dd, doublet of doublet” should be replaced with “dd, doublet of doublets”.

      5)      Line 332. Here and further. Delete the spaces around dash in the numbering intervals like in 7.90 – 7.83, i.e. correct on 7.90–7.83.

      6)      Ref. 17. Correct, please the style of the title as in all other references.

      7)      The main flaw of the article is an absence of the signal assignments in the NMR data. Provide, please, the assignment of all the signals for all the  substances and present them in special tables or by any other appropriate way in Supporting materials.

My opinion: minor revision.

Author Response

Pr. Mohamed Othman

Normandie Univ., UNILEHAVRE, FR 3038 CNRS, URCOM  

76600 Le Havre, France.

EA 3221, INC3M CNRS-FR 3038, UFR ST, BP: 1123,

25 rue Philipe Lebon, F-76063 Le Havre Cedex, France

Phone: (+33) 02.32.74.43.98 ; Office fax: (+33) 02.32.74.43.91

E-mail: mohamed.othman@univ-lehavre.fr; othmanlehavre@gmail.com

Le Havre, 2023 May 30

To          Ms. Miranda Yan
Assistant Editor

            Molecules

Manuscript ID: molecules-2432047

Title: Novel Oleanolic acid-Phtalimidines tethered 1,2,3 triazole hybrids as
promising antibacterial agents: Design, Synthesis, in vitro experiments and
in silico docking studies

Authors: Ghofrane Lahmadi, Mabrouk Horchani, Amal Dbeibia, Abdelkarim Mahdhi,
Anis Romdhane, Ata Martin Lawson, Adam Daïch, Abdel Halim Harrath, Hichem
Ben Jannet *, And Mohamed Othman *

Dear colleague,

We would like to express our deep appreciation for your concern regarding the submitted manuscript. We are thankful for the quick and informative reply from the reviewers. We are also grateful for the valuable comments and suggestions from the reviewers aiming to improve the scientific quality of the submitted manuscript.

After reading the reviewers’ comments/suggestions and discussing their comments with other authors of the manuscript, we revised our manuscript and replied to the comments and questions in a point-by-point fashion. Here we enclose the response to the reviewers’ comments.

We will be grateful if the manuscript can be considered for publication in your prestigious journal, Molecules. We deeply appreciate your kind help and concern.

With my very best regards

Sincerely yours,

Pr. Mohamed Othman

Respond to Reviewers’ comments

General remark of the authors to the referees: In order to facilitate the reviewing process, all the modifications we have made in this new version of the paper, including literature citations, appear in yellow and in magenta of the signal assignments in the NMR data.

Answers to referee 2:

Remark 1> Line 22 and further. The numbering 18a-v is not correct because the substance 18u is absent. The correct numbering is 18a-t, 18v. Fix, please here and in other places.

Answer:  The numbering in the manuscript has been rectified as 18a-u.

Remark 2> Line 26, 192–194, 201 etc. Check and correct the style in Latin names of microorganisms. It should be italic, fix, please in the whole text.

Answer: Cheeked and if necessary modified.

Remark 3> Line 238. Correct the frames at the tables 1 and 2.

AnswerFrames of the tables 1 and 2 have been corrected.

Remark 4> Line 314. “dd, doublet of doublet” should be replaced with “dd, doublet of doublets”.

Answer: Amended.

Remark 5> Line 332. Here and further. Delete the spaces around dash in the numbering intervals like in 7.90 – 7.83, i.e. correct on 7.90–7.83.

Answer: The spaces around dashes have been deleted.

Remark 6> Ref. 17. Correct, please the style of the title as in all other references.

Answer: The style of the title has been modified as requested (Ref. 20 now).

Remark 7> The main flaw of the article is an absence of the signal assignments in the NMR data. Provide, please, the assignment of all the signals for all the substances and present them in special tables or by any other appropriate way in Supporting materials.

Answer: We would like to thank the reviewer for his/her comment. We think that it will be very complicated to give more informations on the attributions of the NMR signals in view of the structural complexity of our molecules. We have already deployed a lot of effort (1HNMR, 13CNMR, dept, 2D.. .) to provide the assignment of the major signals for the substances. According to your wishes, more additional details have been provided, please see the NMR data in (3.2.1. General procedure for the preparation of compounds 18) and Supporting materials.

Reviewer 3 Report

REVIEWER'S REPORT

Manucsript title: Novel Oleanolic acid-Phtalimidines tethered 1,2,3 triazole hy-
2

brids as promising antibacterial agents: Design, Synthesis, invitro experiments and in silico docking studies (Authors: Ghofrane Lahmadi1,2, Mabrouk Horchani2, Amal Dbeibia, Abdelkarim Mahdhi, Anis Romdhane, Ata Martin Lawson, Adam Daïch, Abdel Halim Harrath, Hichem Ben Jannet, and Mohamed Othman).

The manuscript describes the design and synthesis of oleanolic acid ((3-hydroxyolean-12-en-28-oic acid, OA-1)-phtalimidines (isoindo-21 linones) conjugates with triazole moieties, as well as the evaluation of their antibacterial efficacy. Notable actions were defined for several of the substances. Furthermore, docking computations were performed for the most active drugs against the target protein from L. monocytogenes.  The findings supported the experimental data by emphasizing the importance of both hydrogen bonding and hydrophobic interactions with the target protein.

 This article, in my opinion,  fits the standards of the Molecules and might be accepted. However, before it can be published, the manuscript should be carefully proofread and supplemented.

I would  advise changing or modifying several passages in the article's text, in particular:

In introduction. The text (in lines 44-45) might be replaced as "It has also been found that OA-1 and its derivatives have significant antibacterial activity with a wide range of MIC values [11-13]." In lines 55-57, the sentence should be written as "As a result, a lot of study has been done over the past few decades...."

The sentences, in lines 61-62, should be written as " Despite its multiple potential applications, OA-1 has not yet been developed as a medicine due to its instability and limited water solubility." In lines 68-73, the sentence should be written as " Given the rising incidence of multidrug-resistant pathogens caused by the widespread use of antibiotics [46], the intriguing biological activities of isoindolinones and OA-1, and our current research interest in the valorization of agricultural waste into eco-efficient, bioactive products, we present here the synthesis of novel triazole-tethered isoindolinones oleanolic acid hybrids..." In line 80,"...to get a distinct insight..." might be replaced by "...to get a better understanding..." 

The section of the text that describes the assessment of structural properties of the compounds (lines 181-189) should be moved to Materials and Methods.

In Antibacterial activity. The text (in lines 202-205) should be corrected as " In certain circumstances, the activity of this chemical exceeds that of specific derivatives and the reference antibiotic. It was discovered to be more active against S. aureus and P. aeruginosa than all of its dervivatives 18a-v, but only slightly more active against S. typhimurium than 18f, 18k, and 18q."

Proofreading should be done for the Conclusion.

 Employing one standard drug (Tetracycline) as a reference, in my opinion, is insufficient. Typically, several standard drugs are utilized as references when assessing the effects of newly generated substances. 

   Instead of or in addition to docking computation, I recommend to assess the molecular and reactivity indices of the test compounds, and by doing  structure-activity relationship (SAR or QSAR) analysis to reveal indices of the test compounds that may affect their activity. It would be usefull to verify if these compounds follow Lipinski's rule predictive to bioavailability, etc. Such study, in my opinion, would be far more valuable than docking computation. The docking study might be useful as an additional approach after efficiency of the compounds' interaction with the target protein has been experimentally determined. It is exceedingly unlikely that these compounds have a single target. Because the efficacy of the molecules' interactions with the intended target protein has not been experimentally defined in this work, I question the significance of the docking.

The English language, in my opinion, needs only minor correction.

Author Response

Pr. Mohamed Othman

Normandie Univ., UNILEHAVRE, FR 3038 CNRS, URCOM  

76600 Le Havre, France.

EA 3221, INC3M CNRS-FR 3038, UFR ST, BP: 1123,

25 rue Philipe Lebon, F-76063 Le Havre Cedex, France

Phone: (+33) 02.32.74.43.98 ; Office fax: (+33) 02.32.74.43.91

E-mail: mohamed.othman@univ-lehavre.fr; othmanlehavre@gmail.com

Le Havre, 2023 May 30

To          Ms. Miranda Yan
Assistant Editor

            Molecules

Manuscript ID: molecules-2432047

Title: Novel Oleanolic acid-Phtalimidines tethered 1,2,3 triazole hybrids as
promising antibacterial agents: Design, Synthesis, in vitro experiments and
in silico docking studies

Authors: Ghofrane Lahmadi, Mabrouk Horchani, Amal Dbeibia, Abdelkarim Mahdhi,
Anis Romdhane, Ata Martin Lawson, Adam Daïch, Abdel Halim Harrath, Hichem
Ben Jannet *, And Mohamed Othman *

Dear colleague,

We would like to express our deep appreciation for your concern regarding the submitted manuscript. We are thankful for the quick and informative reply from the reviewers. We are also grateful for the valuable comments and suggestions from the reviewers aiming to improve the scientific quality of the submitted manuscript.

After reading the reviewers’ comments/suggestions and discussing their comments with other authors of the manuscript, we revised our manuscript and replied to the comments and questions in a point-by-point fashion. Here we enclose the response to the reviewers’ comments.

We will be grateful if the manuscript can be considered for publication in your prestigious journal, Molecules. We deeply appreciate your kind help and concern.

With my very best regards

Sincerely yours,

Pr. Mohamed Othman

Respond to Reviewers’ comments

General remark of the authors to the referees: In order to facilitate the reviewing process, all the modifications we have made in this new version of the paper, including literature citations, appear in yellow and in magenta of the signal assignments in the NMR data.

Answers to referee 3:

I would  advise changing or modifying several passages in the article's text, in particular:

Remark 1> In introduction. The text (in lines 44-45) might be replaced as "It has also been found that OA-1 and its derivatives have significant antibacterial activity with a wide range of MIC values [11-13]." In lines 55-57, the sentence should be written as "As a result, a lot of study has been done over the past few decades...."

Answer: We would like to thank the reviewer for his/her propositions. Changing of the text has been done

Remark 2> The sentences, in lines 61-62, should be written as " Despite its multiple potential applications, OA-1 has not yet been developed as a medicine due to its instability and limited water solubility."

Answer: Changing of the sentences has been performed,

Remark 2> In lines 68-73, the sentence should be written as " Given the rising incidence of multidrug-resistant pathogens caused by the widespread use of antibiotics [46], the intriguing biological activities of isoindolinones and OA-1, and our current research interest in the valorization of agricultural waste into eco-efficient, bioactive products, we present here the synthesis of novel triazole-tethered isoindolinones oleanolic acid hybrids..." In line 80,"...to get a distinct insight..." might be replaced by "...to get a better understanding..." 

Answer: Changing of the sentences has been performed.

Remark 4> The section of the text that describes the assessment of structural properties of the compounds (lines 181-189) should be moved to Materials and Methods.

Answer: We moved the text to the designated part in the manuscript.

Remark 5> In Antibacterial activity. The text (in lines 202-205) should be corrected as " In certain circumstances, the activity of this chemical exceeds that of specific derivatives and the reference antibiotic. It was discovered to be more active against S. aureus and P. aeruginosa than all of its dervivatives 18a-v, but only slightly more active against S. typhimurium than 18f, 18k, and 18q."

Answer: Correction of the text has been done.

Remark 6> Proofreading should be done for the Conclusion.

      Answer: As requested, the conclusion has been reworded.

 Remark 7> Employing one standard drug (Tetracycline) as a reference, in my opinion, is insufficient. Typically, several standard drugs are utilized as references when assessing the effects of newly generated substances. 

Answer: The antibacterial activity was reexamined and explored through a comparison with another reference drug, Chlorhexidine.

Remark 8>  Instead of or in addition to docking computation, I recommend to assess the molecular and reactivity indices of the test compounds, and by doing  structure-activity relationship (SAR or QSAR) analysis to reveal indices of the test compounds that may affect their activity. It would be usefull to verify if these compounds follow Lipinski's rule predictive to bioavailability, etc. Such study, in my opinion, would be far more valuable than docking computation. The docking study might be useful as an additional approach after efficiency of the compounds' interaction with the target protein has been experimentally determined. It is exceedingly unlikely that these compounds have a single target. Because the efficacy of the molecules' interactions with the intended target protein has not been experimentally defined in this work, I question the significance of the docking.

AnswerDruglikeness studies were not carried out because the focus was on predicting the inhibitory effect of the synthesized compounds on the target strain "Listeria monocytogenes" evaluated in vitro (which gave significant results). In addition, the druglikeness studies including Lipinski did not show good results which explains the effectiveness of the compounds tested, (figure below shows the result of one of our synthesized compounds.)

Round 2

Reviewer 1 Report

The manuscript has been properly revised, thus it can be accepted in the present form.